# Solving Positive Linear Programs with Differential Privacy

**Alina Ene** [1]   **Huy L. Nguyen** [2]   **Ta Duy Nguyen** [1]   **Adrian Vladu** [3]

## Abstract

We study differentially private approximation algorithms for positive linear programs (LPs with nonnegative coefficients and variables), focusing on the fundamental families of packing, covering, and mixed packing-covering formulations. We focus on the high-sensitivity, constraint-private regime of Hsu-Roth-Roughgarden-Ullman (ICALP 2014), where neighboring instances may differ by an arbitrary single constraint, so one cannot hope to approximately satisfy every constraint under privacy. We give private solvers that return approximate solutions while violating only a controlled number of constraints. Our algorithms improve the prior instance-dependent guarantees, and also yield new data-independent bounds that depend only on the dimension. Our techniques involve a dense multiplicative weights update method developed from a regularized dual viewpoint, which we analyze in a way that exploits structure specific to positive LPs.

## 1. Introduction

Positive linear programs (LPs)—most notably packing, covering and mixed packing-covering LPs—play a central role in machine learning and computer science, where they are used to model optimization problems with all non-negative variables and constraints. Packing LPs which arise in problems such as resource usage maximization subject to capacity constraints, have a wide range of applications including ad allocation, bandwidth allocation and fractional matching, while covering LPs seen in problems such as cost minimization subject to covering requirements naturally model facility location relaxations, set cover, etc. Mixed packing-

covering LPs combining both types of constraints appear in network flow control, learning with budget and fair allocation problems. A key reason for the importance of this class of LPs is that they admit fast approximation algorithms, including multiplicative weights update, primal-dual methods, which scale to massive datasets common in modern ML systems.

In many scenarios, linear programs directly involve users' sensitive data such as health or financial information. For example, in a packing LP for bandwidth allocation, each constraint may encode a user's capacity or usage limit while in a covering LP modeling minimum-cost provisioning of services, each user's constraint may represent a personal service requirement. In such cases, solving the LP requires the algorithm designer to pay particular attention to protecting the users' private information. This is where differential privacy proves its usefulness. Differential privacy provides a principled framework for protecting users in this context by ensuring that the output of an LP-solving algorithm does not change significantly when the input LP differs by one constraint.

Research on solving linear programs under differential privacy dates back to Hsu et al. (2014) and has since been extended to broader settings and related problems, including private learning of subspaces and halfspaces Bun et al. (2015); Beimel et al. (2019); Kaplan et al. (2020); Gao & Sheffet (2021); Ben-Eliezer et al. (2022), which are fundamental learning theory problems. A key challenge in privately solving linear programs is an inherent impossibility result: it is not possible to simultaneously guarantee differential privacy and satisfy all constraints. Indeed, the addition or removal of a single constraint can change an LP from feasible to infeasible or vice versa, revealing sensitive information. Consequently, any differentially private algorithm must allow for the violation or removal of some constraints. Characterizing upper bounds on the number of constraints that must be dropped to preserve privacy has therefore become a central question in this line of work, including this paper.

To address this challenge, Hsu et al. (2014) introduces a general framework for approximately solving linear programs under differential privacy. However, in this work, explicit constraint-private algorithms and performance guar-

---

[1]Department of Computer Science, Boston University, USA  [2]Khoury College of Computer Sciences, Northeastern University, USA  [3]CNRS & IRIF, Université Paris Cité, France. Correspondence to: Alina Ene <aene@bu.edu>, Huy Le Nguyen <hu.nguyen@northeastern.edu>, Ta Duy Nguyen <taduy@bu.edu>, Adrian Vladu <vladu@irif.fr>.

*Proceedings of the 43rd International Conference on Machine Learning*, Seoul, South Korea. PMLR 306, 2026. Copyright 2026 by the author(s).

antees are provided only for the fractional set cover problem. Moreover, the resulting bounds are instance-dependent: the number of constraints that must be dropped depends on properties of the problem such as the optimal objective value. More recent works by Kaplan et al. (2024); Ene et al. (2025) develop algorithmic approaches for solving general LPs exactly. These methods achieve data-independent guarantees, but incur high-degree polynomial dependence on the problem dimension, which can be overly pessimistic for structured instances such as positive LPs.

In this work, we revisit the problem of solving linear programs with differential privacy, focusing on the class of positive LPs, including packing, covering and mixed packing-covering LPs. We propose new approximation algorithms for solving these problems as well as new analysis that can exploit their structure to improve the number of constraints that are violated due to privacy.

## 1.1. Our contribution

We focus on constraint-private LPs where neighboring inputs can differ by a single constraint. This is the most challenging setting among the various scenarios introduced by Hsu et al. (2014), for which guarantees for private approximation algorithms are not well understood.

We give constraint-private algorithms that output solutions satisfying all constraints of the LP approximately, except for a bounded number of constraints. We provide upper bounds on the number of constraints that might be violated, distinguishing between two types of guarantee: one that only depends on the problem dimension and one that involves data-dependent properties, specific to the LP given as input.

Our results are stated in the following theorem (precise statements are given in subsequent sections).

**Theorem 1.** *(cf. Theorems 7 and 11) Given the approximation factor $\alpha \in (0, 1)$, there exist $(\varepsilon, \delta)$-differentially private algorithms that output a solution to a packing (covering) LP given by constraints $Ax \leq 1$ (respectively, $Ax \geq 1$) for $A \in \mathbb{R}_{\geq 0}^{n \times d}$. With high probability, the output solution satisfies $Ax \leq 1 + \alpha$ (respectively, $Ax \geq 1 - \alpha$), except for at most $s$ constraints where,*

$$s = \widetilde{O}\left(\frac{(A_{\max}\mathsf{OPT})^{1.5}}{\alpha^2 \varepsilon}\right), \text{ or } s = \widetilde{O}\left(\frac{d^{1.5}}{\alpha^{3.5}\varepsilon}\right),$$

*where $\mathsf{OPT}$ is the optimal objective of the LP and $A_{\max}$ denotes the maximum entry of $A$.*

**Theorem 2.** *(cf. Theorem 12) Given the approximation factor $\alpha \in (0, 1)$, there exist $(\varepsilon, \delta)$-differentially private algorithms that output a solution to a mixed packing-covering LP given by constraints $Px \leq 1, Cx \geq 1$ where $P \in \mathbb{R}_{\geq 0}^{p \times d}, C \in \mathbb{R}_{\geq 0}^{c \times d}$. With high probability, the output solution satisfies $Px \leq 1 + \alpha, Cx \geq 1 - \alpha$, except for at*

*most $s$ constraints where,*

$$s = \widetilde{O}\left(\frac{P_{\max}C_{\max}\sqrt{P_{\max} + C_{\max}}V^{2.5}}{\alpha^{4.5}\varepsilon}\right),$$

$$\text{or } s = \widetilde{O}\left(\frac{d^3}{\alpha^6\varepsilon}\right)$$

*and $V = 1^\top x$ for some feasible solution $x$, $P_{\max}, C_{\max}$ denote the maximum entries of $P$ and $C$, respectively.*

In these theorems statements, the $\widetilde{O}$-notation hides logarithmic factors that depend on $n, d, \delta$, the success probability and the range of the input.

*Remark* 3. Our $d$-dependent bounds from these theorems are most useful in the regime where the number of constraints $n$ is much bigger than the problem dimension $d$. A natural example is the problem of setting the prices (the variables) of some common goods among a large user base to maximize revenue while making sure that most users can meet their needs. Here, the number of users $n$ is vast, while the number of variables $d$ remains small. Another example is the combinatorial public project problem where the goal is to select public projects from a set of $d$ candidates, so as to maximize the total utility of $n$ users where $n \gg d$. This is a discrete problem but an LP relaxation can still yield valuable information such as by rounding.

**Comparison with Hsu et al. (2014).** While Hsu et al. (2014) provide a general framework to solve constraint-private LPs, explicit bounds are only provided for fractional set cover problem for which an explicit private oracle can be constructed. For this problem where $A_{\max} = 1$, the number of violated constraints given by Hsu et al. (2014)'s algorithm is $\widetilde{O}\left(\frac{\mathsf{OPT}^2}{\alpha^2\varepsilon}\right)$. By comparison, for this problem, our algorithm gives a bound of $\widetilde{O}\left(\frac{\mathsf{OPT}^{1.5}}{\alpha^2\varepsilon}\right)$. This is an improvement by a factor $\mathsf{OPT}^{0.5}$. At the same time, our work goes beyond this instance-specific bound and provides an upper bound of $\widetilde{O}\left(\frac{d^{1.5}}{\alpha^{3.5}\varepsilon}\right)$. This bound improves the former when $\mathsf{OPT} \gg \frac{d}{\alpha}$.

**Comparison with Kaplan et al. (2024); Ene et al. (2025).** These works focus on solving general LPs with high precision with differential privacy, i.e, achieving $\log$-dependence instead of polynomial dependence on $\frac{1}{\alpha}$. For general LPs, these solvers have high degree polynomial dependence on the dimension. Specifically, Kaplan et al. (2024) have an upper bound of $\widetilde{O}\left(\frac{d^9}{\varepsilon}\right)$ and the best known bound from Ene et al. (2025) is $\widetilde{O}\left(\frac{d^4}{\varepsilon}\right)$. In the special case when the LP has a strictly positive margin $\rho$, these bounds can be improved to $\widetilde{O}\left(\frac{d^5}{\varepsilon}\operatorname{poly}\log\frac{1}{\rho}\right)$ and $\widetilde{O}\left(\frac{d^2}{\varepsilon}\operatorname{poly}\log\frac{1}{\rho}\right)$, respectively. When we can afford to tolerate a low-precision approximation, which is equivalent to creating an LP with margin $\alpha$,

the algorithm by Ene et al. (2025) achieves an upper bound of $\widetilde{O}\left(\frac{d^2}{\varepsilon}\mathrm{poly}\log\frac{1}{\alpha}\right)$ on the number of violated constraints. By comparison, our data-independent bounds are $\widetilde{O}\left(\frac{d^{1.5}}{\alpha^{3.5}\varepsilon}\right)$ for pure problems and $\widetilde{O}\left(\frac{d^3}{\alpha^6\varepsilon}\right)$ for mixed packing-covering LPs. For pure LPs, our algorithms improve the dependence on the problem dimension, while having a worse bound for the mixed packing-covering LPs. On the other hand, our algorithms also offer instance-specific bound, which can improve these data-independent bounds in certain regimes.

**Our technique.** Our main technique is a newly developed Dense Multiplicative Weights Update algorithm for positive LPs. The prior work by Hsu et al. (2014) also uses a version of Dense Multiplicative Weights Update, relying on the Bregman projection approach by Herbster & Warmuth (2001) as a blackbox. However, this blackbox approach fails to leverage the positivity of the constraints. Instead, we develop our new toolset from first principles, which allow us to exploit the structure of the problem.

Our algorithm assigns and update weights to the constraints in the LP and makes sure that these weights do not exceed a certain threshold in order to preserve the privacy when a constraint of the LP changes (thresholding guarantees a bound on the sensitivity). We develop this algorithm from a regularized dual viewpoint, using a variant of the standard softmax function, where the dual variable has its coordinates bounded by a specified threshold. More precisely, the main tool is the truncated softmax function, defined as follows:

$$\mathrm{smax}^U(x) = \max_{r \in \mathcal{D}^U}\langle x, r\rangle - \omega(r),$$

where the domain of the dual variable $\mathcal{D}^U = \{r \in \Delta_n : \max r \leq U\}$ is the unit simplex with coordinates truncated at some value $0 < U < 1$ and $\omega$ is the negative entropy. The weights of the constraints are given by $\nabla\mathrm{smax}^U(x)$. The main idea when using $\mathrm{smax}^U(x)$ is that bounding the regularization term can provide an improved bound on the sensitivity of the LP. Properties of this function and its gradient allow us control the contribution of any set of $s = \frac{1}{U}$ constraints, which exactly provides the guarantee on the number of violated constraints in all cases.

### 1.2. Related Work

Hsu et al. (2014) were the first to initiate the study of solving linear programs with differential privacy. In their work, multiple definitions of neighboring inputs are considered. We only consider the most challenging case of constraint-private LPs where neighboring inputs can differ by a single constraint, which is the same setting studied by Kaplan et al. (2024); Ene et al. (2025). The main tool developed by Hsu et al. (2014) is private MWU to approximately solve LPs. Previously, private MWUs have been developed in the context of linear queries release (Hardt & Rothblum, 2010) to achieve optimal accuracy and runtime. For solving LPs, the aim, on the other hand, is to find a solution that can minimize the number of violated constraints. Kaplan et al. (2024); Ene et al. (2025) use a different approach based on privatizing rescaled perceptron algorithms to solve general LPs to high accuracy.

Our work focus on the class of positive LPs. Where privacy is not a concern, there has been a long line of research developing multiplicative weights update for positive linear programs to improve runtime over algorithms for general LPs, starting from Shahrokhi & Matula (1990); Plotkin et al. (1995); Grigoriadis & Khachiyan (1994); Luby & Nisan (1993); Young (2001) followed by many subsequent works (see the survey by Arora et al. (2012)). It is therefore natural to seek analogous improvement when privacy is imposed. Our work show that improvement is indeed possible.

## 2. Preliminaries

### 2.1. Notation

For a matrix $A$, we use $A_{\min}$ and $A_{\max}$ to denote the minimum and maximum entries of $A$. We use $A_i$ to denote the $i$-th row. We use a scalar $a \in \mathbb{R}$ to denote the vector whose coordinates are all $a$. The dimension can be inferred from the context.

### 2.2. Positive Linear Programs

We consider the problem of solving linear programs with positive entries. There are three main families of such LPs.

**Packing LPs**. We seek to approximately solve the problem

$$\max_{x \geq 0} c^\top x \text{ s.t } Ax \leq b,$$

where $A \in \mathbb{R}^{n \times d}_{\geq 0}, b \in \mathbb{R}^n_{\geq 0}, c \in \mathbb{R}^d_{\geq 0}$. Without loss of generality, we can assume that $b = 1$ and $c = 1$.

**Covering LPs**. We seek to approximately solve the problem

$$\min_{x \geq 0} c^\top x \text{ s.t } Ax \geq b,$$

where $A \in \mathbb{R}^{n \times d}_{\geq 0}, b \in \mathbb{R}^n_{\geq 0}, c \in \mathbb{R}^d_{\geq 0}$. We will also assume that $b = 1$ and $c = 1$.

**Mixed Packing-Covering LPs**. This is the most general positive LPs, for which we want to find $x \geq 0$ such that

$$Px \leq b \text{ and } Cx \geq c,$$

where $P \in \mathbb{R}^{p \times d}_{\geq 0}, C \in \mathbb{R}^{c \times d}_{\geq 0}, b \in \mathbb{R}^p_{\geq 0}, c \in \mathbb{R}^c_{\geq 0}$. We will assume that $b = 1$ and $c = 1$.

## 2.3. Differential Privacy

We will represent an LP by its constraint matrix, after normalization so that the objective coefficients and scalars are 1s. We say that two LPs with constraint matrices $A$ and $A'$ are neighbors if they differ by a single constraint. A randomized algorithm $\mathcal{A}$ is said to be $(\varepsilon, \delta)$-differentially private (DP) if for all neighboring LPs $A$ and $A'$ and every subset of possible outcomes $\mathcal{O}$,

$$\Pr\left[\mathcal{A}(A) \in \mathcal{O}\right] \leq e^\varepsilon \Pr\left[\mathcal{A}(A') \in \mathcal{O}\right] + \delta.$$

In the case $\delta = 0$, we say the algorithm is $\varepsilon$-DP.

We will use the exponential mechanism (McSherry & Talwar, 2007)—a standard tool in differential privacy. The exponential mechanism involves a score function $Q(i, A)$ which takes as input a candidate $i$ from a finite set $S$ and a dataset $A$ and outputs a real value. Given a dataset $A$, the exponential mechanism $\mathsf{EM}_{\varepsilon, Q}$ outputs $i \in S$ with probability proportional to $\exp\left(\varepsilon Q(i, A)\right)$.

We have the following guarantee.

**Theorem 4.** *The exponential mechanism is $\varepsilon$-differentially private and guarantee that for a score function $Q$ with sensitivity $\Delta$, with probability at least $1 - \beta$,*

$$Q(\mathsf{EM}_{\varepsilon, Q}(A), A) \geq \max_{j \in S} Q(j, S) - \frac{2\Delta\left(\log d + \log \frac{1}{\beta}\right)}{\varepsilon},$$

*where we say a score function $Q$ has sensitivity $\Delta$ if for all $i \in S$ and any neighboring inputs $A$ and $A'$, $|Q(i, A) - Q(i, A')| \leq \Delta$.*

## 2.4. Tool: Truncated Softmax and Softmin

The main tool we use to analyze our algorithm is the truncated softmax function, defined as follows. Let $\Delta_n = \left\{r \in \mathbb{R}^n_{\geq 0} : \sum_i r_i = 1\right\}$ be the unit simplex. Given a scalar parameter $0 < U \leq 1$, define $\mathcal{D}^U = \left\{r \in \Delta_n : \max r \leq U\right\}$. We define

$$\mathrm{smax}^U(x) = \max_{r \in \mathcal{D}^U} \langle x, r \rangle - \omega(r)$$

where $\omega(r) = \sum_i r_i \log r_i$ is the negative entropy. We also define the truncated softmin function $\mathrm{smin}^U$ as

$$\mathrm{smin}^U(x) = -\mathrm{smax}^U(-x) = \min_{r \in \mathcal{D}^U} \langle x, r \rangle + \omega(r).$$

**Gradient.** The gradient of $\mathrm{smax}^U(x)$ is given by

$$\left[\nabla \mathrm{smax}^U(x)\right]_i = \min\left\{U, \exp\left(x_i - t_U(x)\right)\right\}$$

where $t_U(x)$ is such that $\nabla \mathrm{smax}^U(x) \in \Delta_n$ i.e. $\sum_i \left[\nabla \mathrm{smax}^U(x)\right]_i = 1$. Similarly

$$\left[\nabla \mathrm{smin}^U(x)\right]_i = \min\left\{U, \exp\left(-x_i - t_U(x)\right)\right\}$$

where $t_U(x)$ is such that $\nabla \mathrm{smin}^U(x) \in \Delta_n$ i.e. $\sum_i \left[\nabla \mathrm{smin}^U(x)\right]_i = 1$.

**Bounding the $\mathrm{smax}^U$ and $\mathrm{smin}^U$ increase.** We bound the change in $\mathrm{smax}^U$ when performing an update $x \leftarrow x + u$ for a vector $u \in \mathbb{R}^n_{\geq 0}$.

**Proposition 5.** *For all $x$ and $u \in \mathbb{R}^n_{\geq 0}$*

$$\mathrm{smax}^U(x + u) \leq \mathrm{smax}^U(x) + \frac{e^D - 1}{D}\left\langle \nabla \mathrm{smax}^U(x), u \right\rangle,$$

*where $D = \max_i \{u_i\}$.*

*Proof.* Let $g(t) = \mathrm{smax}^U(x + tu)$. Also let $C(t)$ be the set of coordinates $i$ for which $\left[\nabla \mathrm{smax}^U(x + tu)\right]_i = U$ and $\overline{C(t)} = [n] \setminus C(t)$. We have

$$g'(t) = \left\langle \nabla \mathrm{smax}^U(x + tu), u \right\rangle$$
$$g''(t) = u^\top \nabla^2 \mathrm{smax}^U(x + tu)\, u$$
$$\overset{(*)}{\leq} u^\top \mathrm{diag}\left(\left[\begin{array}{c} \nabla \mathrm{smax}^U(x + tu)\Big|_{\overline{C(t)}} \\ 0_{C(t)} \end{array}\right]\right) u$$
$$\leq D \cdot \left\langle \nabla \mathrm{smax}^U(x + tu), u \right\rangle = D \cdot g'(t),$$

where in $(*)$ we use Proposition 14. Thus $g''(t) \leq D \cdot g'(t)$. Note that $g'$ is continuous (recall that since we regularize with a strongly convex regularizer, $\mathrm{smax}^U$ is smooth and thus $\nabla \mathrm{smax}^U$ is continuous) and the inequality holds on each of the intervals where $C(t)$ is constant. Thus $\log g'(t)$ is also continuous and

$$\frac{d}{dt} \log g'(t) = \frac{g''(t)}{g'(t)} \leq D,$$

and therefore

$$\log g'(t) \leq \log g'(0) + t \cdot D$$
$$g'(t) \leq \exp(t \cdot D) \cdot g'(0)$$

and so we have

$$g(1) = g(0) + \int_0^1 g'(t)\, dt$$
$$\leq g(0) + \int_0^1 \exp(t \cdot D)\, g'(0)\, dt$$
$$= g(0) + g'(0) \cdot \frac{e^D - 1}{D}.$$

Substituting the value of $g$, we obtain the claim. $\square$

From Proposition 5, in particular, for $u \in \mathbb{R}^n_{\geq 0}$ such that $\max_i \{u_i\} \leq 1$, we also have

$$\mathrm{smax}^U(x + u) \leq \mathrm{smax}^U(x) + (1 + D)\left\langle \nabla \mathrm{smax}^U(x), u \right\rangle. \tag{1}$$

Similarly,

$$\mathrm{smin}^U(x + u) \geq \mathrm{smin}^U(x) + (1 - D)\left\langle \nabla \mathrm{smin}^U(x), u \right\rangle. \tag{2}$$

## 3. Private Algorithms for Packing LP

In this section, we seek to approximately solve the following problem with differential privacy

$$\max_{x \geq 0} 1^\top x \text{ s.t } Ax \leq 1,$$

where $A \in \mathbb{R}_{\geq 0}^{n \times d}$. We assume to know the value $\mathsf{OPT} = \max\{1^\top x, Ax \leq 1\}$, so the goal is, given the approximation factor $\alpha$, to find $x$ such that

$$1^\top x \geq (1 - \alpha)\,\mathsf{OPT} \text{ and } Ax \leq 1 + \alpha.$$

We also assume the entries of the input matrix $A$ come from a bounded range and are upper bounded by a known value $S = O(A_{\max})$.

*Remark* 6. The assumption that we know the value of OPT is also made in Hsu et al. (2014) and implicitly in Kaplan et al. (2024); Ene et al. (2025). Here, the implicit assumption is that the maximum coefficient for each variable $x_j$ across all constraints lies within a range $[m, M]$ for known values $M > m > 0$. One then has $\mathsf{OPT} \in [\frac{1}{dM}, \frac{d}{m}]$. We can run the algorithm for each multiple of $(1 + \alpha)$ in this range which covers a $(1 \pm \alpha)$ approximation of OPT. Then using the exponential mechanism, we can select the best guess of OPT with only $\log$ factors incurred. By the same reason, using a binary search in $[m, M]$, we can estimate the value $S = O(A_{\max})$. We will ignore these $\log$ factors due to the binary searches in our bounds.

### 3.1. Algorithm

Our algorithm is presented in Figure 1. Starting with the initial solution $x = 0$, he algorithm proceeds iteratively as follows. In each iteration, the algorithm calls PackingOracle, which uses the exponential mechanism to find a coordinate $j$ such that $\langle \nabla \mathrm{smax}^U(\eta Ax), A1_j \rangle$ is minimized to form the update vector $\Delta_t$. The output of the algorithm is the average of the cumulative updates, i.e, $\overline{x} = \frac{\sum_t \Delta_t}{T}$.

Algorithm 1 is closely related to the Dense Multiplicative Weights Update algorithm by Herbster & Warmuth (2001) which is employed by Hsu et al. (2014) as a blackbox in their private algorithm for solving LPs. One key distinction here is that our Dense MWU algorithm exploits the structure of the positive LPs in maintaining that in each update, the increase in the constraint values $\mathrm{smax}^U(\eta A(x_{t+1})) - \mathrm{smax}^U(\eta Ax_t)$ is small when updating $x_{t+1} = x_t + \Delta_t$ with an appropriate step size $\eta$. More specifically, without the constraint positivity, Hsu et al. (2014) bound

$$\mathrm{smax}^U(\eta A(x_{t+1})) - \mathrm{smax}^U(\eta Ax_t)$$
$$\lesssim \langle \nabla \mathrm{smax}^U(\eta Ax_t), \eta A\Delta_t \rangle + \eta^2 \|A\Delta_t\|_\infty^2.$$

---

**Algorithm 1** Private Algorithm for Packing LP

1: **Input**: $A \in \mathbb{R}^{n \times d}$, upper bound $S = O(A_{\max})$ for the entries of $A$, optimal objective $\mathsf{OPT} = \max\{1^\top x, Ax \leq 1\}$, $\alpha, \beta \in (0, 1)$, privacy parameters $\varepsilon, \delta$.

2: Let $H = \frac{2d}{\alpha \cdot \mathsf{OPT}}$ if pre-processing else $H = S$

3: Let $T = \frac{20H \cdot \mathsf{OPT} \log n}{\alpha^2}$, $\varepsilon' = \frac{\varepsilon}{2\sqrt{T \log \frac{1}{\delta}}}$, $s = \frac{60H \cdot \mathsf{OPT}(\log d + \log \frac{T}{\beta})}{\alpha \varepsilon'}$, $U = \frac{1}{s}$

4: **(Optional) Pre-processing**: for entry $A_{ij}$ in $A$: $A_{ij} = \min\{A_{ij}, H\}$

5: Initialize $x_0 = 0$

6: for $t = 0, \ldots, T - 1$:

7:     Let $\Delta_t = \mathsf{PackingOracle}(x_t, \varepsilon')$

8:     $x_{t+1} = x_t + \Delta_t$

9: Let $\overline{x} = \frac{x_T}{T}$

10: **(Optional) Post-processing**: for $j \in [d]$: if $\overline{x}_j \leq \frac{2}{H}$, let $\overline{x}_j = 0$

11: **Output** $\overline{x}$

12: PackingOracle$(x, \varepsilon)$:

13:     Let $\eta = \frac{\alpha}{10H \cdot \mathsf{OPT}}$

14:     Use the Exponential mechanism with privacy parameter $\varepsilon$ to find coordinate $j \in [d]$ with score function $Q(j) = -\langle \nabla \mathrm{smax}^U(\eta Ax), A1_j \rangle \mathsf{OPT}$

15:     **Output** $\Delta = 1_j \cdot \mathsf{OPT}$

---

The term $\|A\Delta_t\|_\infty$ results in the convergence rate $\frac{\rho^2 \log n}{\alpha^2}$ of Dense MWU for general LP, where $\rho$ is the width of the problem, being an upper bound for $\|A\Delta_t\|_\infty$. On the other hand, if we have the constraint positivity, we can bound

$$\mathrm{smax}^U(\eta A(x_{t+1})) - \mathrm{smax}^U(\eta Ax_t)$$
$$\lesssim (1 + \eta\|A\Delta_t\|_\infty)\langle \nabla \mathrm{smax}^U(\eta Ax_t), \eta A\Delta_t \rangle,$$

and improve the convergence rate to $\frac{\rho \log n}{\alpha^2}$. The improvement of the runtime of the non-private algorithm by a factor $\rho$ leads to the improvement in the number of violated constraints in our private algorithm. Intuitively, every iteration requires some privacy loss so having fewer iterations allows us to utilize the privacy budget more efficiently.

At the same time, in each iteration, PackingOracle guarantees the objective increases quickly ($1^\top \Delta_t = \mathsf{OPT}$) and the average of the constraints $Ax \leq 1$ weighted by $\nabla \mathrm{smax}^U(\eta Ax_{t-1})$, i.e $\langle \nabla \mathrm{smax}^U(\eta Ax_{t-1}), A\Delta_t \rangle$ is satisfied approximately. Here, $\nabla \mathrm{smax}^U(\eta Ax_{t-1})$ can be thought of as the projection of weights $\nabla \mathrm{smax}(\eta Ax_{t-1})$ onto a dense distribution over the simplex $\Delta_n$ where no weights exceed $U$ in order to achieve privacy. The use of the truncated softmax function $\mathrm{smax}^U$ allows us to control the sensitivity of the score function $Q$ used in the exponential mechanism, which directly translates to the number of constraints that have to be dropped. This function also gives

the bound

$$\max_{S\in[n]:|S|=s=\frac{1}{U}} \left\langle \frac{1_S}{|S|}, A\overline{x} \right\rangle \leq \text{smax}^U (A\overline{x})$$

which implies if $\overline{x}$ satisfies $\text{smax}^U (A\overline{x}) \leq 1 + \alpha$, the average of the top $\frac{1}{U}$ constraints cannot exceed $(1 + \alpha)$ and thus the number of violated constraints is at most $\frac{1}{U}$.

It is optional to execute pre-processing and post-processing steps to obtain the novel data-independent guarantee for the algorithm. In the pre-processing step, the algorithm truncates the entries of the input matrix $A$ by threshold $H = \frac{2d}{\alpha\cdot\text{OPT}}$. This step makes sure that we can have an instance-independent bound for the sensitivity of the score function used in the exponential mechanism. To make sure that the output solution satisfies the original constraints instead of the new constraints resulted from the pre-processing, the entries of $\overline{x}$ that are below the threshold $\frac{\alpha\text{OPT}}{d}$ are set to 0.

The guarantee of Algorithm 1 is stated below.

**Theorem 7.** *Given $\alpha, \beta \in (0,1)$, Algorithm 1 is $(\varepsilon, \delta)$-differentially private and finds a solution $x$ such that $1^\top x \geq (1 - \alpha)\,\text{OPT}$ and with probability at least $1 - \beta$, $A_i x \leq 1 + \alpha$ for all $i \in [n]$ except for at most $s$ constraints, where*

*1. Without pre- and post-processing:*

$$s = O\left( \frac{(A_{\max}\text{OPT})^{1.5} (\log d + \log \frac{n}{\alpha\beta}) \sqrt{\log n \log \frac{1}{\delta}}}{\alpha^2 \varepsilon} \right).$$

*2. With pre- and post-processing:*

$$s = O\left( \frac{d^{1.5}(\log d + \log \frac{n}{\alpha\beta}) \sqrt{\log n \log \frac{1}{\delta}}}{\alpha^{3.5} \varepsilon} \right).$$

### 3.2. Proof of Theorem 7

**Privacy guarantee.** We have the following lemma.

**Lemma 8.** *Algorithm 1 is $(\varepsilon, \delta)$-DP.*

*Proof.* In each iteration, we use the Exponential mechanism with privacy parameter $\varepsilon'$ to find $\Delta_t$. By strong composition over $T$ iterations, Algorithm 1 is $\left( \varepsilon' \sqrt{2T \log \frac{1}{\delta}} + \frac{T\varepsilon'^2}{2}, \delta \right)$-DP. For $\varepsilon' = \frac{\varepsilon}{2\sqrt{T \log \frac{1}{\delta}}}$ and constant $\varepsilon$, this implies Algorithm 1 is $(\varepsilon, \delta)$-DP. □

**Utility Analysis.** First, we have the following guarantee of PackingOracle.

**Lemma 9.** *With probability at least $1 - \beta$, for all $t \in [T]$:*

$$\langle \nabla smax^U (\eta A x_{t-1}), A\Delta_t \rangle \leq 1 + \frac{\alpha}{10}.$$

*Proof.* We calculate the sensitivity of the score function $Q(j) = -\left\langle \nabla\text{smax}^U (\eta Ax), A1_j \right\rangle \text{OPT}$. As a reminder, we let $H$ be the upper bound on the entries of $A$. If the algorithm does not do pre-processing, we simply have $H = S$. Otherwise, $H = \frac{2d}{\alpha\cdot\text{OPT}}$.

Note that the coordinate of $\nabla\text{smax}^U$ is bounded by $U = \frac{1}{s}$ and the entries of $A$ are bounded by $\frac{2d}{\alpha\cdot\text{OPT}}$. For any $x$ and any neighboring inputs $A$ and $A'$ (after the preproccesion),

$$\left| \left\langle \nabla\text{smax}^U (\eta Ax), A1_j \right\rangle - \left\langle \nabla\text{smax}^U (\eta A'x), A'1_j \right\rangle \right|$$
$$\leq \left| \left\langle \nabla\text{smax}^U (\eta Ax), A1_j \right\rangle - \left\langle \nabla\text{smax}^U (\eta Ax), A'1_j \right\rangle \right|$$
$$+ \left| \left\langle \nabla\text{smax}^U (\eta Ax), A'1_j \right\rangle - \left\langle \nabla\text{smax}^U (\eta A'x), A'1_j \right\rangle \right|$$
$$\leq \frac{3H}{s}.$$

Hence the sensitivity of $Q$ is $\frac{3H\cdot\text{OPT}}{s}$. Since there is a solution $x^*$ which satisfies $1^\top x^* = \text{OPT}$ and $Ax^* \leq 1$, there must exist a coordinate $j$ such that $\left\langle \nabla\text{smax}^U (\eta Ax_{t-1}), A1_j \right\rangle \text{OPT} \leq 1$. There are $d$ coordinates, so with probability $1 - \frac{\beta}{T}$, in each iteration, the Exponential mechanism guarantees

$$\langle \nabla\text{smax}^U (\eta Ax_{t-1}), A\Delta_t \rangle$$
$$\leq 1 + \frac{6H \cdot \text{OPT} \left( \log d + \log \frac{T}{\beta} \right)}{s\varepsilon'}.$$

Since $s = \frac{60H\cdot\text{OPT}\left(\log d + \log \frac{T}{\beta}\right)}{\alpha\varepsilon'}$, we have $\langle \nabla\text{smax}^U (\eta Ax_{t-1}), A\Delta_t \rangle \leq 1 + \frac{\alpha}{10}$. The lemma statement can be obtained by taking the union bound over $T$ iterations. □

**Lemma 10.** *With probability $1 - \beta$, the output $\overline{x}$ of Algorithm 1 satisfies $1^\top \overline{x} \geq (1 - \alpha)\,\text{OPT}$ and $A_i \overline{x} \leq 1 + \alpha$ except for at most $s$ constraints where*

*1. Without pre- and post-processing:*

$$s = O\left( \frac{(A_{\max}\text{OPT})^{1.5} (\log d + \log \frac{n}{\alpha\beta}) \sqrt{\log n \log \frac{1}{\delta}}}{\alpha^2 \varepsilon} \right).$$

*2. With pre- and post-processing:*

$$s = O\left( \frac{d^{1.5}(\log d + \log \frac{n}{\alpha\beta}) \sqrt{\log n \log \frac{1}{\delta}}}{\alpha^{3.5} \varepsilon} \right).$$

*Proof.* First, we show that $A_i \overline{x} \leq 1 + \alpha$ except for at most $s$ constraints. We have $\|A\Delta_t\|_\infty \leq \max_{i,j} A_{ij} \cdot \text{OPT} \leq H \cdot \text{OPT}$. Hence, for $\eta = \frac{\alpha}{10H\cdot\text{OPT}}$ and for all $t$, $\|\eta A\Delta_t\|_\infty \leq$

$\frac{\alpha}{10} \leq 1$. Using property (1) of the $\text{smax}^U$ function

$$\text{smax}^U \left(\eta A x_T\right) \leq \text{smax}^U \left(\eta A x_{T-1}\right)$$
$$+ \left(1 + \|\eta A \Delta_t\|_\infty\right) \left\langle \nabla \text{smax}^U \left(\eta A x_{T-1}\right), \eta A \Delta_T \right\rangle$$
$$\leq \text{smax}^U \left(\eta A x_{T-1}\right)$$
$$+ \left(1 + \frac{\alpha}{10}\right) \left\langle \nabla \text{smax}^U \left(\eta A x_{T-1}\right), \eta A \Delta_T \right\rangle.$$

Unrolling this over $T$ iterations, we obtain

$$\text{smax}^U \left(\eta A x_T\right) \leq \text{smax}^U \left(0\right)$$
$$+ \eta \left(1 + \frac{\alpha}{10}\right) \sum_{t=1}^{T-1} \left\langle \nabla \text{smax}^U \left(\eta A x_t\right), A \Delta_{t+1} \right\rangle$$
$$\overset{(i)}{\leq} \log n + \eta \left(1 + \frac{\alpha}{10}\right) \cdot T \cdot \left(1 + \frac{\alpha}{10}\right) \overset{(ii)}{\leq} \eta T \left(1 + \alpha\right).$$

where $(i)$ comes from Lemma 9 and $(ii)$ is due to the choice $\eta = \frac{\alpha}{10 H \cdot \text{OPT}}$ and $T = \frac{2 \log n}{\eta \alpha}$. We then have

$$\max_{S \in [n] : |S| = s = \frac{1}{U}} \left\langle \frac{1_S}{|S|}, \eta A x_T \right\rangle \leq \max_{r \in \Delta_n, r \leq U} \left\langle r, \eta A x_T \right\rangle - \omega\left(r\right)$$
$$= \text{smax}^U \left(\eta A x_T\right) \leq \eta T \left(1 + \alpha\right).$$

Therefore, we have that

$$\max_{S \in [n] : |S| = s} \left\langle \frac{1_S}{|S|}, A \overline{x} \right\rangle \leq \max_{S \in [n] : |S| = s} \left\langle \frac{1_S}{|S|}, A \frac{x_T}{T} \right\rangle \leq 1 + \alpha.$$

This means among $s$ constraints with the largest values $A_i \overline{x}$, the smallest value is upperbounded by $1 + \alpha$. This implies all but at most $s$ constraints $A_i x \leq 1 + \alpha$ are satisfied, where

$$s = O \left( \frac{H \cdot \text{OPT} \left(\log d + \log \frac{T}{\beta}\right)}{\alpha \varepsilon'} \right).$$

**Without pre- and post-processing**. We have

$$s = O \left( \frac{S \cdot \text{OPT} \left(\log d + \log \frac{T}{\beta}\right)}{\alpha \varepsilon'} \right)$$
$$= O \left( \frac{\left(A_{\max} \text{OPT}\right)^{1.5} \left(\log d + \log \frac{n}{\alpha \beta}\right) \sqrt{\log n \log \frac{1}{\delta}}}{\alpha^2 \varepsilon} \right),$$

where $S = O\left(A_{\max}\right)$. Further, since $\Delta_t = 1_j \text{OPT}$ for some coordinate $j$, we have $1^\top \Delta_t = \text{OPT}$. Hence, $\overline{x}$ satisfies: $1^\top \overline{x} = \frac{1}{T} \sum_t 1^\top \Delta_t = \text{OPT}$, as needed.

**With pre- and post-processing**. We have

$$s = O \left( \frac{d \left(\log d + \log \frac{T}{\beta}\right)}{\alpha^2 \varepsilon'} \right)$$
$$= O \left( \frac{d^{1.5} \left(\log d + \log \frac{n}{\alpha \beta}\right) \sqrt{\log n \log \frac{1}{\delta}}}{\alpha^{3.5} \varepsilon} \right).$$

For the objective, we also have, $\frac{1}{T} \sum_t 1^\top \Delta_t = \text{OPT}$. The output of the algorithm $\overline{x}$ is obtained by truncating the entries of $\frac{1}{T} \sum_t 1^\top \Delta_t$ that are smaller than $\frac{\alpha \text{OPT}}{d}$. Therefore

$$1^\top x \geq \text{OPT} - d \cdot \frac{\alpha \text{OPT}}{d} = \left(1 - \alpha\right) \text{OPT}.$$

Next we show that at most $s$ constraints before truncation are not satisfied. To distinguish between the input before and after the truncation, let us denote by $A^{\text{init}}$ the original input and $A$ the truncated input. Let $i \in [n]$ be such that $A_i \overline{x} \leq 1 + \alpha$. For $j \in [d]$ such that $A_{ij}^{\text{init}} \leq \frac{2d}{\alpha \cdot \text{OPT}}$, we have $A_{ij}^{\text{init}} = A_{ij}$, so $A_{ij}^{\text{init}} \overline{x}_j = A_{ij} \overline{x}_j$. For $j \in [d]$ such that $A_{ij}^{\text{init}} > \frac{2d}{\alpha \cdot \text{OPT}}$, we have $A_{ij} = \frac{2d}{\alpha \cdot \text{OPT}}$. It follows that $\overline{x}_j \leq \frac{1 + \alpha}{A_{ij}} \leq \frac{\alpha \text{OPT}}{d}$, which means $\overline{x}_j = 0$. We then have $A_{ij}^{\text{init}} \overline{x}_j = 0 \leq A_{ij} \overline{x}_j$. Overall, $A_i^{\text{init}} \overline{x} \leq A_i \overline{x} \leq 1 + \alpha$. Since at most $s$ truncated constraints are violated, it follows that at most $s$ original constraints are violated. $\square$

## 4. Private Algorithm for Covering LP

In this section, we seek to approximately solve the following problem with differential privacy

$$\min 1^\top x \text{ s.t } A x \geq 1, x \geq 0,$$

where $A \in \mathbb{R}_{\geq 0}^{n \times d}$. We assume to know the value $\text{OPT} = \min \left\{1^\top x, A x \geq 1\right\}$, so the goal is, given the approximation factor $\alpha$, to find $x$ such that

$$1^\top x \leq \left(1 + \alpha\right) \text{OPT} \text{ and } A x \geq 1 - \alpha.$$

We also assume the entries of the input matrix $A$ are upper bounded by a known value $R = O\left(A_{\max}\right)$, which can be found with a binary search with only $\log$-dependence on the range of the entries of $A$.

### 4.1. Algorithm

Our algorithm is presented in Figure 2. Similarly to Algorithm 1, we also start with an optional preprocessing step where large entries of the input matrix $A$ are truncated by threshold $H = \frac{40d}{\alpha \cdot \text{OPT}}$. This step is required only when we want to obtain a data-independent guarantee. Next, the algorithm proceeds iteratively. In each iteration, the algorithm calls CoveringOracle, which uses the Exponential mechanism to find an update vector $\Delta_t$. This vector $\Delta_t$ satisfies $1^\top \Delta_t = \text{OPT}$ and with high probability $\left\langle \nabla \text{smin}^U \left(\eta A x_{t-1}\right), A \Delta_t \right\rangle \geq 1 - O\left(\alpha\right)$, with the parameters defined in the algorithm. The output of the algorithm is the average of the cumulative updates, i.e, $\overline{x} = \frac{\sum_t \Delta_t}{T}$.

The guarantee of Algorithm 2 is stated below. The proof follows similarly to that of Theorem 7 which we defer to Appendix B.

---

**Algorithm 2** Private Algorithm for Covering LP

---

**Input**: $A \in \mathbb{R}^{n \times d}$, upper bound $R = O(A_{\max})$ for the entries of $A$, optimal objective $\mathsf{OPT} = \min\{1^\top x, Ax \geq 1\}$, $\alpha, \beta \in (0,1)$, privacy parameters $\varepsilon, \delta$.

1: Let $H = \frac{40d}{\alpha \cdot \mathsf{OPT}}$ if pre-processing else $H = R$
2: **(Optional) Pre-processing**: for entry $A_{ij}$ in $A$: $A_{ij} = \min\{A_{ij}, H\}$
3: Let $T = \frac{20H \cdot \mathsf{OPT} \log n}{\alpha}$, $\varepsilon' = \frac{\varepsilon}{2\sqrt{T \log \frac{1}{\delta}}}$, $s = \frac{120H \cdot \mathsf{OPT}\left(\log d + \log \frac{T}{\beta}\right)}{\alpha \varepsilon'}$, $U = \frac{1}{s}$
4: Initialize $x_0 = 0$
5: **for** $t = 0, \ldots, T-1$:
6:      Let $\Delta_t = \mathsf{CoveringOracle}(x_t, \varepsilon')$
7:      $x_{t+1} = x_t + \Delta_t$
8: **Output** $\overline{x} = \frac{x_T}{T}$
9: $\mathsf{CoveringOracle}(x, \varepsilon)$:
10:      Let $\eta = \frac{\alpha}{10H \cdot \mathsf{OPT}}$
11:      Use the Exponential mechanism with privacy parameter $\varepsilon$ to find coordinate $j \in [d]$ with score function $Q(i) = \left\langle \nabla \mathrm{smin}^U(\eta A x), A 1_j \right\rangle \mathsf{OPT}$
12:      **Output** $\Delta = 1_j \mathsf{OPT}$

---

**Theorem 11.** *Given $\alpha, \beta \in (0,1)$, Algorithm 2 is $(\varepsilon, \delta)$-differentially private and finds a solution $x$ such that $1^\top x \leq (1+\alpha)\mathsf{OPT}$ and with probability at least $1 - \beta$, $A_i x \geq 1 - \alpha$ for all $i \in [n]$ except for at most $s$ constraints, where*

*1. Without pre-processing:*

$$s = O\left(\frac{(A_{\max}\mathsf{OPT})^{1.5} \log d \log \frac{n}{\alpha\beta} \sqrt{\log n \log \frac{1}{\delta}}}{\alpha^2 \varepsilon}\right).$$

*2. With pre-processing:*

$$s = O\left(\frac{d^{1.5} \log d \log \frac{n}{\alpha\beta} \sqrt{\log n \log \frac{1}{\delta}}}{\alpha^{3.5} \varepsilon}\right).$$

# 5. Mixed Packing-Covering LP with DP

In this section, we seek to approximately solve the following problem with differential privacy: Find $x \geq 0$ such that

$$Px \leq 1 \text{ and } Cx \geq 1,$$

where $P \in \mathbb{R}_{\geq 0}^{p \times d}$, $C \in \mathbb{R}_{\geq 0}^{c \times d}$. The goal is, given the approximation factor $\alpha$, to find $x$ such that

$$Px \leq 1 + \alpha \text{ and } Cx \geq 1 - \alpha.$$

Similar to the previous cases, by Remark 6, we assume that the maximum entry in each column of $P$ and $C$ come from a bounded range $[m, M]$ for known values $M > m > 0$. By

a binary search, we can find upper bounds $S$ for the entries of $P$ and $R$ for the entries of $C$ such that $S = O(P_{\max})$ and $R = O(C_{\max})$. We also assume to know a value $V = 1^\top x$ for a feasible solution $x$.

We give two algorithms to solve this problem: one with guarantee depending on the input data and one with guarantee only depending on the problem dimensions.

**Theorem 12.** *Given $\alpha, \beta \in (0,1)$ and let $n = p + c$. Assuming that the maximum entry in each column of $P$ and $C$ lie in range $[m, M]$ for known values $M > m > 0$, we have the following*

*1. Algorithm 3 is $(\varepsilon, \delta)$-differentially private and finds a solution $x$ such that with probability at least $1 - \beta$, $P_i x \leq 1 + \alpha$ for all $i \in [p]$ and $C_i x \geq 1 - \alpha$ for all $i \in [c]$, except for at most $s$ constraints, where*

$$s = O\left(\frac{P_{\max}C_{\max}\sqrt{P_{\max} + C_{\max}} V^{2.5}}{\alpha^{4.5}\varepsilon}\right.$$
$$\left. \cdot \left(\log d + \log \frac{n}{\alpha\beta}\right)\sqrt{\log \frac{1}{\delta} \log n}\right),$$

*and $V = 1^\top x$ for some feasible solution $x$ of the LP.*

*2. Algorithm 4 is $(\varepsilon, \delta)$-differentially private and finds a solution $x$ such that with probability at least $1 - \beta$, $P_i x \leq 1 + \alpha$ for all $i \in [p]$ and $C_i x \geq 1 - \alpha$ for all $i \in [c]$, except for at most $s$ constraints, where*

$$s = O\left(\frac{d^3 \left(\log d + \log \frac{n}{\alpha\beta}\right)\sqrt{\log \frac{1}{\delta} \log n}}{\alpha^6 \varepsilon}\right.$$
$$\left. + \frac{d^2 \left(\log\log \frac{M}{m} + \log \frac{d}{\beta}\right)\log \frac{M}{m}}{\varepsilon}\right).$$

The higher order of dependence on the problem parameters/dimension and the approximation factor of both algorithms compared with the counterparts for pure LPs is expected, due to the more complex interplay between the packing and covering constraints. Even when privacy is not considered, mixed packing-covering LPs require more iterations for MWU algorithms to find an approximate solution.

## 5.1. Algorithm with data-dependent guarantee

Algorithm 3 uses $\mathsf{MPCOracle}$ to find the coordinate $j$ so that the relative increase between the packing and covering constraints, characterized by $\frac{\left\langle \nabla \mathrm{smax}^U(Px), P 1_j \right\rangle}{\left\langle \nabla \mathrm{smin}^U(Cx), C 1_j \right\rangle}$ is minimized (i.e, we want the increase of the packing constraints is small compared with that of the covering constraints). Similarly to the cases of pure LPs, we use the exponential mechanism to find this coordinate privately with score

**Algorithm 3** Mixed Packing-Covering LP

1: **Input**: $P \in \mathbb{R}_{\geq 0}^{p \times d}$, $C \in \mathbb{R}_{\geq 0}^{c \times d}$, $\alpha, \beta \in (0,1)$, privacy parameters $\varepsilon, \delta$, $V = 1^\top x$ for some solution $x$
2: Initialize $x_0 = 0$
3: Let $C_{ij} \leftarrow C_{ij} + \frac{\alpha}{V}$ for all $i, j$
4: Let $T = \Theta\left((S+R)V\frac{\log n}{\alpha^3}\right)$, $\varepsilon' = \frac{\varepsilon}{2\sqrt{T \log \frac{1}{\delta}}}$,
5: Let $s = O\left(\frac{SR\sqrt{S+R}V^{2.5}(\log d + \log \frac{T}{\beta})\sqrt{\log \frac{1}{\delta}}\log n}{\alpha^{4.5}\varepsilon}\right)$,
   $U = \frac{1}{s}$
6: for $t = 0, \ldots, T-1$:
7:    Let $\Delta_t = \mathsf{MPCOracle}(x_t, \varepsilon')$
8:    $x_{t+1} = x_t + \Delta_t$
9: Use the Exponential mechanism with privacy parameter $\varepsilon'$ to find $k$ such that $k \cdot x_T$ minimizes the number of violated constraints for $k$ being power of $1 + \alpha$ in range $[\frac{m}{\alpha TM}, \frac{60M}{\alpha m}]$.
10: **Output** $\overline{x} = k \cdot x_T$
11: $\mathsf{MPCOracle}(x, \varepsilon)$:
12:    Use the Exponential mechanism with privacy parameter $\varepsilon$ to find coordinate $j \in [d]$ with score function
   $$Q(j) = -\frac{\langle \nabla\mathrm{smax}^U(Px), P1_j\rangle}{\langle \nabla\mathrm{smin}^U(Cx), C1_j\rangle}$$
13:    **Output** $\Delta = \frac{\alpha \cdot 1_j}{30(S+R)}$.

function $Q(j) = -\frac{\langle \nabla\mathrm{smax}^U(Px), P1_j\rangle}{\langle \nabla\mathrm{smin}^U(Cx), C1_j\rangle}$. However, the sensitivity of this score function can be large since the term $\langle \nabla\mathrm{smin}^U(Cx), C1_j\rangle$ can be as small as $C_{\min}$. To avoid the case when $C_{\min}$ is close to 0, we perturb the entries of $C$ by adding to them $\frac{\alpha}{V}$. While maintaining that $1^\top x = V$, the additional term $\frac{\alpha}{V} \cdot 1^\top x$ only contributes $\alpha$ to the covering constraints, hence if the modified covering constraints are satisfied, the original are also satisfied approximately.

Note that $x_T$ guarantees that the ratio $\frac{\mathrm{smax}^U(Px_T)}{\mathrm{smin}^U(Cx_T)} \leq 1 + \alpha$. A final step before outputting the cumulated solution $x_T$ is to find the scale $k$ so that $\overline{x} = k \cdot x_T$ so that the constraints are satisfied. This can be done by using the exponential mechanism in combination with a binary search.

We defer the analysis to Appendix C.

**5.2. Algorithm with data-independent guarantee**

To obtain a bound independent of the input data, we need to pre-process the data. As in the case of pure covering LPs, we need to clip the large entries of the covering matrix. The challenge here is that the clipping threshold has to take the packing constraints into account and needs to be relative to the entries of the packing constraints. For this reason, we need to estimate the maximum entries for each coordinate of the packing matrix. This can be done relatively easy using the exponential mechanism over the range of the entries

$[m, M]$. Once we have the estimate of $M_j = \max_i\{P_{ij}\}$, we can clip the entries of the covering matrix by $\frac{40dM_j}{\alpha}$, besides perturbing them by $\frac{\alpha M_j}{d}$ as in Algorithm 3. The algorithm then proceeds iteratively, similarly to Algorithm 3. We give the full analysis in Appendix D.

**Algorithm 4** Mixed Packing-Covering LP

1: **Input**: $P \in \mathbb{R}_{\geq 0}^{p \times d}$, $C \in \mathbb{R}_{\geq 0}^{c \times d}$, $\alpha, \beta \in (0,1)$, privacy parameters $\varepsilon, \delta$, known range $[m, M]$ for $\max_j\{P_{ij}\}$, $\max_j\{C_{ij}\}$.
2: **Pre-processing**:
3:    for $j \in [d]$:
4:       Estimate $M_j = \max_i\{P_{ij}\}$ and filter out $i$ such that $P_{ij} \geq M_j$ using $\mathsf{MaxEstimator}\left(m, M, \{P_{ij}\}_{j=1}^p, \frac{\varepsilon}{2d}, \frac{\beta}{2d}\right)$
5:       Let $C_{ij} \leftarrow \min\left\{C_{ij} + \frac{\alpha M_j}{d}, \frac{40dM_j}{\alpha}\right\}$, for all $i, j$
6: Initialize $x_0 = 0$,
7: Let $T = \Theta\left(\frac{d^2 \log n}{\alpha^4}\right)$, $\varepsilon' = \frac{\varepsilon}{4\sqrt{T \log \frac{1}{\delta}}}$,
8: Let $s = O\left(\frac{d^3(\log d + \log \frac{T}{\beta})\sqrt{\log \frac{1}{\delta}}\log n}{\alpha^6 \varepsilon}\right)$, $U = \frac{1}{s}$
9: for $t = 0, \ldots, T-1$:
10:    Let $\Delta_t = \mathsf{MPCOracle}(x_t, \varepsilon')$
11:    $x_{t+1} = x_t + \Delta_t$
12: Use the Exponential mechanism with privacy parameter $\varepsilon'$ to find $k$ such that $k \cdot x_T$ minimizes the number of violated constraints for $k$ being power of $1 + \alpha$ in range $[\frac{m}{\alpha TM}, \frac{60M}{\alpha m}]$.
13: **Output** $\overline{x} = k \cdot x_T$
14: $\mathsf{MaxEstimator}(m, M, \{a_i\}, \varepsilon, \beta)$:
15:    Use the Exponential mechanism with privacy parameter $\varepsilon$ to find $k$ such that $2^k m \in [m, 2M]$ with score function $Q(k) = -\left|\{a_i : a_i \geq 2^k m\}\right| - \frac{2\left(\log\log\frac{2M}{m} + \log\frac{1}{\beta}\right)}{\varepsilon}k$
16:    **Output** $2^k m$.

## Acknowledgments

HLN was supported in part by NSF grant CCF-2311649 and a gift from Apple Inc. AV was partially supported by the French Agence Nationale de la Recherche (ANR) under grant ANR-21-CE48-0016 (project COMCOPT).

## Impact Statement

This paper presents work whose goal is to advance the field of machine learning. There are many potential societal consequences of our work, none of which we feel must be specifically highlighted here.

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

# A. Tools: Truncated Softmax and Softmin

We expand the properties of $\mathrm{smax}^U$ and $\mathrm{smin}^U$ and their proofs from Section 2. Recall that, given a scalar parameter $0 < U \leq 1$, define $\mathcal{D}^U = \{r \in \Delta_n : \max r \leq U\}$. We define

$$\mathrm{smax}^U(x) = \max_{r \in \mathcal{D}^U} \langle x, r \rangle - \omega(r)$$

where $\omega(r) = \sum_i r_i \log r_i$ is the negative entropy.

**Proposition 13.** *The gradient of $\mathrm{smax}^U(x)$ is given by*

$$\left[\nabla \mathrm{smax}^U(x)\right]_i = \min\{U, \exp(x_i - t_U(x))\}.$$

*Proof.* Using Lagrange multipliers we write

$$\max_{r \in \mathcal{D}^U} \langle x, r \rangle - \omega(r) = \max_r \min_{y \geq 0, z \geq 0, \lambda} \langle x, r \rangle - \omega(r) + \langle y, r \rangle + \langle z, U - r \rangle + \lambda(1 - \langle 1, r \rangle)$$

$$= \min_{y \geq 0, z \geq 0, \lambda} \max_r \langle x + y - z - \lambda 1, r \rangle - \omega(r) + \langle z, U \rangle + \lambda.$$

By first order optimality for the maximization problem we have

$$x + y - z - \lambda 1 = 1 + \log r$$

and satisfy complementary slackness conditions $z_i(U - r)_i = 0$, $y_i r_i = 0$.

We also note that by complementary slackness $y$ is nonzero only in the degenerate case where all the entries of $x$ are either $U$ or 0. Indeed, let $C = \{i : r_i = U\}$, and let $\alpha = 1 - U \cdot |C|$ denote the remaining mass not on maximized entries. Provided that $\alpha > 0$, we can verify via a standard argument that $r$ will put mass on all coordinates, otherwise it contradicts optimality. Hence we only have to consider

$$r = \exp(x - z - t)$$

for some scalar $t$, and where $z_i(U - r)_i = 0$. This means that entries of $r$ smaller than $U$ satisfy $z_i = 0$ and hence this entries satisfy $r_i = \exp(x_i - t)$. Otherwise $r_i = U$. This leads to the expression

$$\left[\nabla \mathrm{smax}^U(x)\right]_i = r_i \in \{U, \exp(x_i - t_U(x))\}$$

with $t_U(x)$ chosen so that the resulting vector belongs to the unit simplex. $\qquad \square$

**Proposition 14.** *Let $C = \left\{i : \left[\nabla \mathrm{smax}^U(x)\right]_i = U\right\}$ and $\overline{C} = [n] \setminus C$. We have*

$$\nabla^2 \mathrm{smax}^U(x) \preceq diag\left(\begin{bmatrix} \left.\nabla \mathrm{smax}^U(x)\right|_{\overline{C}} \\ 0_C \end{bmatrix}\right).$$

*Proof.* It is easy to verify that $\nabla^2 \mathrm{smax}^U(x)$ must be 0 on rows/columns corresponding to entries in $C$. Meanwhile all the entries in $\left.\nabla \mathrm{smax}^U\right|_{\overline{C}}$ have $\alpha = 1 - U \cdot |C|$ total mass distributed among them. Hence we can view $\left.\nabla \mathrm{smax}^U\right|_{\overline{C}}$ as a vector defined over $\alpha \Delta_{|\overline{C}|}$. In this case we can verify that the gradient is just $g = \alpha \cdot \frac{\exp(x)}{\sum \exp(x)}$ and thus the Hessian satisfies the standard identity $H = \alpha \cdot \left(\mathrm{diag}(g) - gg^\top\right)$. Thus we have that

$$\nabla^2 \mathrm{smax}^U(x) = \mathrm{diag}\left(\begin{bmatrix} \left.\nabla \mathrm{smax}^U(x)\right|_{\overline{C}} \\ 0_C \end{bmatrix}\right) - \frac{1}{\alpha} \cdot \begin{bmatrix} \left.\nabla \mathrm{smax}^U(x)\right|_{\overline{C}} \\ 0_C \end{bmatrix} \begin{bmatrix} \left.\nabla \mathrm{smax}^U(x)\right|_{\overline{C}} \\ 0_C \end{bmatrix}^\top$$

$$\preceq \mathrm{diag}\left(\begin{bmatrix} \left.\nabla \mathrm{smax}^U(x)\right|_{\overline{C}} \\ 0_C \end{bmatrix}\right).$$

$\qquad \square$

## B. Analysis of Algorithm 2

**Privacy guarantee.**

**Lemma 15.** *Algorithm 2 is $(\varepsilon, \delta)$-DP.*

*Proof.* The proof is similar to the packing case. □

**Utility Analysis.** To show the number of constraints dropped, first, we have the following guarantee of CoveringOracle.

**Lemma 16.** *With probability at least $1 - \beta$, for all $t \in [T]$:*

$$\left\langle \nabla smin^U \left( \eta A x_{t-1} \right), A \Delta_t \right\rangle \geq 1 - \frac{\alpha}{10}.$$

To prove this lemma, we use the following claim which shows the change in the LP if the algorithm truncates the input entries. Again, let us denote by $A^{\mathrm{init}}$ and $A$ the original and the truncated input matrices for distinction.

*Claim 17.* There exists $y \geq 0$ such that $1^\top y = \mathsf{OPT}$ and $Ay \geq 1 - \frac{\alpha}{20}$.

*Proof.* Let $x^*$ be such that $1^\top x^* = \mathsf{OPT}$ and $A^{\mathrm{init}} x \geq 1$. Let $y = \frac{1}{1 + \alpha/20} \left( x^* + \frac{\alpha \mathsf{OPT}}{20d} \right)$. We have $1^\top y = \mathsf{OPT}$. Consider constraint $i$. If there is some $j$ such that $A_{ij}^{\mathrm{init}} \geq \frac{40d}{\alpha \mathsf{OPT}}$, we have $A_i y \geq \frac{40d}{\alpha \mathsf{OPT}} \cdot \frac{\alpha \mathsf{OPT}}{20d(1+\alpha)} \geq 1$. Otherwise we have $A_i = A_i^{\mathrm{init}}$. Hence

$$A_i y = A_i^{\mathrm{init}} y \geq A_i^{\mathrm{init}} \frac{x^*}{1 + \alpha/20} \geq \frac{1}{1 + \alpha/20} \geq 1 - \frac{\alpha}{20}.$$

Therefore there exists $y \geq 0$ such that $1^\top y = \mathsf{OPT}$ and $Ay \geq 1 - \frac{\alpha}{20}$. □

*Proof.* We calculate the sensitivity of the score function $Q(i) = \left\langle \nabla smin^U \left( \eta A x \right), A 1_i \right\rangle \mathsf{OPT}$. Note that the coordinate of $\nabla smax^U$ is upperbounded by $U = \frac{1}{s}$ and the entries of $A$ are bounded by $H$. For any $x$ and any neighboring inputs $A$ and $A'$ (after prepoccessing),

$$\begin{aligned}
&\left| \left\langle \nabla smin^U \left( \eta A x \right), A 1_i \right\rangle - \left\langle \nabla smin^U \left( \eta A' x \right), A' 1_i \right\rangle \right| \\
\leq\ & \left| \left\langle \nabla smin^U \left( \eta A x \right), A 1_i \right\rangle - \left\langle \nabla smin^U \left( \eta A x \right), A' 1_i \right\rangle \right| \\
& + \left| \left\langle \nabla smin^U \left( \eta A x \right), A' 1_i \right\rangle - \left\langle \nabla smin^U \left( \eta A' x \right), A' 1_i \right\rangle \right| \\
\leq\ & \frac{3H}{s}.
\end{aligned}$$

Hence the sensitivity of $Q$ is $\frac{3H}{s}$. By Claim 17, there must exist a coordinate $j$ such that $A 1_j \mathsf{OPT} \geq 1 - \frac{\alpha}{20}$. There are $d$ coordinates, so with probability $1 - \frac{\beta}{T}$, in each iteration, the Exponential mechanism guarantees

$$\begin{aligned}
\left\langle \nabla smin^U \left( \eta A x_{t-1} \right), A \Delta_t \right\rangle &\geq 1 - \frac{\alpha}{20} - \frac{6H \cdot \mathsf{OPT} \left( \log d + \log \frac{T}{\beta} \right)}{s \varepsilon'} \\
&\geq 1 - \frac{\alpha}{10},
\end{aligned}$$

where $s = \frac{120 H \cdot \mathsf{OPT} \left( \log d + \log \frac{T}{\beta} \right)}{\alpha \varepsilon'}$. The lemma statement can be obtained by taking the union bound over $T$ iterations. □

**Lemma 18.** *With probability $1 - \beta$, the output $\bar{x}$ of Algorithm 2 satisfies $1^\top \bar{x} = \mathsf{OPT}$ and $A_i \bar{x} \leq 1 + \alpha$ except for at most $s$ constraints where*

*1. Without pre-processing:*

$$s = O \left( \frac{\left( A_{\max} \mathsf{OPT} \right)^{1.5} \left( \log d + \log \frac{n}{\alpha \beta} \right) \sqrt{\log n \log \frac{1}{\delta}}}{\alpha^2 \varepsilon} \right).$$

*2. With pre-processing:*

$$s = O\left(\frac{d^{1.5}\left(\log d + \log \frac{n}{\alpha\beta}\right)\sqrt{\log n \log \frac{1}{\delta}}}{\alpha^{3.5}\varepsilon}\right).$$

*Proof.* Since $\Delta_t = 1_j \mathsf{OPT}$ for some coordinate $j$, we have $1^\top \Delta_t = \mathsf{OPT}$. Hence,

$$1^\top \overline{x} = \frac{1}{T}\sum_t 1^\top \Delta_t = \mathsf{OPT}.$$

Next we show that at most $s$ constraints are not satisfied. To distinguish between the input before and after the truncation, let us denote by $A^{\mathrm{init}}$ the original input and $A$ the truncated input. Since $A_i^{\mathrm{init}} \geq A_i$ for all $i$, we only need to show that $A_i\overline{x} \geq 1 - \alpha$ except for at most $s$ constraints.

We have $\|A\Delta_t\|_\infty \leq \max_{i,j} A_{ij} \cdot \mathsf{OPT} \leq H \cdot \mathsf{OPT}$. Hence, for $\eta = \frac{\alpha}{10H \cdot \mathsf{OPT}}$ and for all $t$, $\|\eta A\Delta_t\|_\infty \leq \frac{\alpha}{10} \leq 1$. Using property (1) of the $\mathrm{smax}^U$ function

$$\mathrm{smin}^U(\eta Ax_T) \geq \mathrm{smin}^U(\eta Ax_{T-1}) + (1 - \|\eta A\Delta_t\|_\infty)\langle\nabla\mathrm{smin}^U(\eta Ax_{T-1}), \eta A\Delta_T\rangle$$
$$\geq \mathrm{smin}^U(\eta Ax_{T-1}) + \left(1 - \frac{\alpha}{10}\right)\langle\nabla\mathrm{smin}^U(\eta Ax_{T-1}), \eta A\Delta_T\rangle.$$

Unrolling this over $T$ iterations, we obtain

$$\mathrm{smin}^U(\eta Ax_T) \geq \mathrm{smin}^U(0) + \eta\left(1 - \frac{\alpha}{10}\right)\sum_{t=1}^{T-1}\langle\nabla\mathrm{smin}^U(\eta Ax_t), A\Delta_{t+1}\rangle$$
$$\overset{(i)}{\geq} -\log n + \eta\left(1 - \frac{\alpha}{10}\right)\cdot T \cdot \left(1 - \frac{\alpha}{10}\right)$$
$$\overset{(ii)}{\geq} \eta T(1 - \alpha).$$

where $(i)$ comes from Lemma 16 and $(ii)$ is due to the choice $\eta = \frac{\alpha}{10H \cdot \mathsf{OPT}}$ and $T = \frac{2\log n}{\eta\alpha}$.

We then have

$$\min_{S\in[n]:|S|=s=\frac{1}{U}}\left\langle\frac{1_S}{|S|}, \eta Ax_T\right\rangle \geq \min_{r\in\Delta_n, r\leq U}\langle r, \eta Ax_T\rangle + \omega(r)$$
$$= \mathrm{smin}^U(\eta Ax_T)$$
$$\geq \eta T(1 - \alpha).$$

Therefore, we have that

$$\min_{S\in[n]:|S|=s}\left\langle\frac{1_S}{|S|}, A\overline{x}\right\rangle = \min_{S\in[n]:|S|=s}\left\langle\frac{1_S}{|S|}, A\frac{x_T}{T}\right\rangle \geq 1 - \alpha.$$

This means among the top $s$ constraints with the smallest values $A_i\overline{x}$, the largest value of them is lower bounded by $1 - \alpha$. This implies all but at most $s$ constraints $A_ix \geq 1 - \alpha$ are satisfied.

**Without pre-processing:**

$$s = O\left(\frac{R \cdot \mathsf{OPT}\left(\log d + \log\frac{T}{\beta}\right)}{\alpha\varepsilon'}\right)$$
$$= O\left(\frac{(A_{\max}\mathsf{OPT})^{1.5}\left(\log d + \log\frac{n}{\alpha\beta}\right)\sqrt{\log n \log\frac{1}{\delta}}}{\alpha^2\varepsilon}\right).$$

**With pre-processing**:

$$s = O\left(\frac{d\left(\log d + \log \frac{T}{\beta}\right)}{\alpha^2 \varepsilon'}\right)$$

$$= O\left(\frac{d^{1.5}\left(\log d + \log \frac{n}{\alpha\beta}\right)\sqrt{\log n \log \frac{1}{\delta}}}{\alpha^{3.5}\varepsilon}\right).$$

$\square$

## C. Analysis of Algorithm 3

**Privacy guarantee.**

**Lemma 19.** *Algorithm 3 is* $(\varepsilon, \delta)$-*DP.*

*Proof.* The proof is similar to the packing case. $\square$

**Utility Analysis.** To show the number of constraints dropped, first, we have the following guarantee of MPCOracle.

**Lemma 20.** *With probability at least* $1 - \beta$, *for all* $t \in [T]$:

$$\frac{\left\langle \nabla smax^U \left(Ax_{t-1}\right), P\Delta_t\right\rangle}{\left\langle \nabla smin^U \left(Ax_{t-1}\right), C\Delta_t\right\rangle} \le 1 + \frac{\alpha}{5}.$$

*Proof.* We calculate the sensitivity of the score function $Q(j) = -\frac{\left\langle \nabla smax^U (Px), P1_j\right\rangle}{\left\langle \nabla smin^U (Cx), C1_j\right\rangle}$. We consider two cases: when the neighboring inputs differ by a covering constraint and when the differ by a packing constraint.

Case 1: Suppose that neighboring inputs are given by $(P, C)$ and $(P, C')$. Note that after pre-processing, we have $C_{\min}$ and $C'_{\min}$ are bounded from below by $\alpha/V$. Thus we have

$$\left| \frac{\left\langle \nabla smax^U (Px), P1_j\right\rangle}{\left\langle \nabla smin^U (Cx), C1_j\right\rangle} - \frac{\left\langle \nabla smax^U (Px), P1_j\right\rangle}{\left\langle \nabla smin^U (C'x), C'1_j\right\rangle}\right|$$

$$= \left|\left\langle \nabla smax^U (Px), P1_j\right\rangle\right| \cdot \frac{\left|\left\langle \nabla smin^U (Cx), C1_j\right\rangle - \left\langle \nabla smin^U (C'x), C'1_j\right\rangle\right|}{\left\langle \nabla smin^U (Cx), C1_j\right\rangle\left\langle \nabla smin^U (C'x), C'1_j\right\rangle}$$

$$\le S\frac{3R/s}{C_{\min}C'_{\min}} \le \frac{3SRV^2}{s\alpha^2}.$$

Case 2: Suppose that neighboring inputs are given by $(P, C)$ and $(P', C)$. We also have

$$\left| \frac{\left\langle \nabla smax^U (Px), P1_j\right\rangle}{\left\langle \nabla smin^U (Cx), C1_j\right\rangle} - \frac{\left\langle \nabla smax^U (P'x), P'1_j\right\rangle}{\left\langle \nabla smin^U (Cx), C1_j\right\rangle}\right|$$

$$\le \frac{3S}{sC_{\min}} \le \frac{3SRV^2}{s\alpha^2}.$$

There exists a coordinate $j$ such that

$$\frac{\left\langle \nabla smax^U \left(Px_{t-1}\right), P1_j\right\rangle}{\left\langle \nabla smin^U \left(Cx_{t-1}\right), C1_j\right\rangle} \le 1.$$

Therefore with probability at least $1 - \frac{\beta}{T}$,

$$\frac{\left\langle \nabla \operatorname{smax}^U (Px_{t-1}), P1_j \right\rangle}{\left\langle \nabla \operatorname{smin}^U (Cx_{t-1}), C1_j \right\rangle} \leq 1 + \frac{6SRV^2 (\log d + \log \frac{T}{\beta})}{s\alpha^2 \varepsilon'}$$

$$\leq 1 + \frac{\alpha}{5},$$

by the choice of $s$.  $\square$

**Lemma 21.** *With probability $1 - \beta$, the output $\bar{x}$ of Algorithm 3 satisfies $P_i \bar{x} \leq 1 + \alpha$ and $C_i \bar{x} \geq 1 - \alpha$ except for at most $s$ constraints where*

$$s = O\left( \frac{P_{\max} C_{\max} \sqrt{P_{\max} + C_{\max}} V^{2.5}}{\alpha^{4.5} \varepsilon} \right.$$

$$\left. \cdot \left( \log d + \log \frac{n}{\alpha \beta} \right) \sqrt{\log \frac{1}{\delta} \log n} \right).$$

*Proof.* By the update $\Delta_t = \frac{\alpha \cdot 1_i}{30(P_{\max} + C_{\max})}$, we have $\|C\Delta_t\|_\infty \leq \frac{\alpha}{30} \leq 1$, hence,

$$\operatorname{smin}^U (Cx_T) \geq \operatorname{smin}^U (Cx_{T-1}) + (1 - \|C\Delta_t\|_\infty) \left\langle \nabla \operatorname{smin}^U (Cx_{T-1}), C\Delta_T \right\rangle$$

$$\geq \operatorname{smin}^U (Cx_{T-1}) + \left(1 - \frac{\alpha}{30}\right) \left\langle \nabla \operatorname{smin}^U (Cx_{T-1}), C\Delta_T \right\rangle$$

$$\geq \operatorname{smin}^U (0) + \left(1 - \frac{\alpha}{4}\right) \sum_{t=1}^{T-1} \left\langle \nabla \operatorname{smin}^U (Cx_t), C\Delta_{t+1} \right\rangle$$

Similarly, we have $\|P\Delta_t\|_\infty \leq \frac{\alpha}{30} \leq 1$, hence,

$$\operatorname{smax}^U (Px_T) \leq \operatorname{smax}^U (Px_{T-1}) + (1 + \|P\Delta_t\|_\infty) \left\langle \nabla \operatorname{smax}^U (Px_{T-1}), P\Delta_T \right\rangle$$

$$\leq \operatorname{smax}^U (Px_{T-1}) + \left(1 + \frac{\alpha}{30}\right) \left\langle \nabla \operatorname{smax}^U (Px_{T-1}), P\Delta_T \right\rangle$$

$$\leq \operatorname{smax}^U (Px_{T-1}) + \left(1 + \frac{\alpha}{30}\right) \left(1 + \frac{\alpha}{5}\right) \left\langle \nabla \operatorname{smin}^U (Cx_{T-1}), C\Delta_T \right\rangle$$

$$\leq \operatorname{smax}^U (0) + \left(1 + \frac{\alpha}{4}\right) \sum_{t=1}^{T-1} \left\langle \nabla \operatorname{smin}^U (Cx_t), C\Delta_{t+1} \right\rangle$$

where in the third inequality, we use Lemma 20. Combining both inequalities, we have

$$\frac{\operatorname{smax}^U (Px_T)}{\operatorname{smin}^U (Cx_T)} \leq \frac{\log n + \left(1 + \frac{\alpha}{4}\right) \sum_{t=1}^{T-1} \left\langle \nabla \operatorname{smin}^U (Cx_t), C\Delta_{t+1} \right\rangle}{-\log n + \left(1 - \frac{\alpha}{4}\right) \sum_{t=1}^{T-1} \left\langle \nabla \operatorname{smin}^U (Cx_t), C\Delta_{t+1} \right\rangle}$$

$$= \frac{\frac{2}{\left(1 - \frac{\alpha}{4}\right)} \log n - \frac{\left(1 + \frac{\alpha}{4}\right)}{\left(1 - \frac{\alpha}{4}\right)} \log n + \left(1 + \frac{\alpha}{4}\right) \sum_{t=1}^{T-1} \left\langle \nabla \operatorname{smin}^U (Cx_t), C\Delta_{t+1} \right\rangle}{-\log n + \left(1 - \frac{\alpha}{4}\right) \sum_{t=1}^{T-1} \left\langle \nabla \operatorname{smin}^U (Cx_t), C\Delta_{t+1} \right\rangle}$$

$$= \frac{\frac{2}{\left(1 - \frac{\alpha}{4}\right)} \log n}{-\log n + \left(1 - \frac{\alpha}{4}\right) \sum_{t=1}^{T-1} \left\langle \nabla \operatorname{smin}^U (Cx_t), C\Delta_{t+1} \right\rangle} + \frac{\left(1 + \frac{\alpha}{4}\right)}{\left(1 - \frac{\alpha}{4}\right)}.$$

Note that we set $\Delta_t = \frac{\alpha}{30(P_{\max} + C_{\max})} \cdot 1_i$, we have

$$\sum_{t=1}^{T-1} \left\langle \nabla \operatorname{smin}^U (Cx_t), C\Delta_{t+1} \right\rangle \geq \frac{\alpha C_{\min} \cdot T}{30(P_{\max} + C_{\max})}$$

By the choice $T = \frac{P_{\max} + C_{\max}}{C_{\min}} \frac{480 \log n}{\alpha^2}$ we also have

$$\log n \leq \frac{\alpha}{8} \left( -\log n + \left( 1 - \frac{\alpha}{4} \right) \frac{\alpha C_{\min} \cdot T}{30 (P_{\max} + C_{\max})} \right)$$

This implies

$$\frac{\text{smax}^U (P x_T)}{\text{smin}^U (C x_T)} \leq 1 + \alpha.$$

It follows that

$$\frac{\max_{S \in [p] : |S| = s = \frac{1}{U}} \left\langle \frac{1_S}{|S|}, P x_T \right\rangle}{\min_{S' \in [c] : |S'| = s = \frac{1}{U}} \left\langle \frac{1_{S'}}{|S'|}, C x_T \right\rangle} \leq \frac{\text{smax}^U (P x_T)}{\text{smin}^U (C x_T)} \leq 1 + \alpha.$$

This means that for all $i \in [p] \setminus S$ and $k \in [c] \setminus S'$ $x_T$ satisfied $\frac{P_i x_T}{C_k x_T} \leq 1 + \alpha$. If we scale $\overline{x} = k x_T$ such that $\max_i P_i \overline{x} = 1$, we $C_i \overline{x} \geq 1 - \alpha$, except for at most $s$ constraint. Since

$$\frac{\alpha m}{60 M} \leq \max_i P_i x_T \leq \frac{\alpha T M}{30 m}$$

therefore

$$k \in \left[ \frac{m}{\alpha T M}, \frac{60 M}{\alpha m} \right]$$

Using the exponential mechanism for $(1 + \alpha)$ powers to minimize the number of violated constraints, we have the number of violated constraints is upper bounded by

$$s = O \left( \frac{SRV^2 \log d \log \frac{T}{\beta}}{\alpha^3 \varepsilon'} \right) = O \left( \frac{SRV^2 (\log d + \log \frac{T}{\beta}) \sqrt{T \log \frac{1}{\delta}}}{\alpha^3 \varepsilon} \right)$$

$$= O \left( \frac{P_{\max} C_{\max} \sqrt{P_{\max} + C_{\max}} V^{2.5} (\log d + \log \frac{n}{\alpha \beta}) \sqrt{\log \frac{1}{\delta} \log n}}{\alpha^{4.5} \varepsilon} \right).$$

$\square$

## D. Analysis of Algorithm 4

First, we examine the guarantee of $\mathsf{MaxEstimator}(m, M, \{a_i\}, \varepsilon, \beta)$.

**Lemma 22.** *Given $m, M, \varepsilon, \beta$, with probability at least $1 - \beta$, $\mathsf{MaxEstimator}(m, M, \{a_i\}, \varepsilon, \beta)$ outputs $K = 2^k m$ such that $K \leq 4 \max_i \{a_i\}$ and $|\{a_i : a_i > K\}| \leq O \left( \frac{(\log \log \frac{M}{m} + \log \frac{1}{\beta}) \log \frac{M}{m}}{\varepsilon} \right)$.*

*Proof.* Let $k^*$ be the smallest number such that $2^k m \geq \max \{a_i\}$. We have $2^{k^*} m < 2 \max \{a_i\}$. We have $Q$ has sensitivity 1 and $\max_{k : 2^k m \in [m, 2M]} Q(k) \geq -\frac{2 \left( \log \log \frac{2M}{m} + \log \frac{1}{\beta} \right)}{\varepsilon} k^*$. The guarantee of the exponential mechanism gives us: with probability $\geq 1 - \beta$,

$$Q(k) \geq -\frac{2 \left( \log \log \frac{2M}{m} + \log \frac{1}{\beta} \right)}{\varepsilon} (k^* + 1).$$

Therefore $k \leq k^* + 1$ and $\left| \{a_i : a_i > 2^k m\} \right| \leq \frac{2 \left( \log \log \frac{2M}{m} + \log \frac{1}{\beta} \right)}{\varepsilon} (k^* + 1) \leq \frac{2 \left( \log \log \frac{2M}{m} + \log \frac{1}{\beta} \right) \left( \log \frac{2M}{m} + 1 \right)}{\varepsilon}$ $\square$

**Lemma 23.** *The Pre-processing step in Algorithm is $\frac{\varepsilon}{2}$-DP and with probability at least $1 - \frac{\beta}{2}$, the number of filtered out constraints is bounded by*

$$O\left(\frac{d^2\left(\log\log\frac{M}{m} + \log\frac{d}{\beta}\right)\log\frac{M}{m}}{\varepsilon}\right).$$

*Proof.* The privacy guarantee is obtained by composition of $d$ calls to MaxEstimator, each of which is $\frac{\varepsilon}{2d}$-DP. The number of filtered constraints by each coordinate is at most $O\left(\frac{d\left(\log\log\frac{M}{m} + \log\frac{d}{\beta}\right)\log\frac{M}{m}}{\varepsilon}\right)$ with probability $\geq 1 - \frac{\beta}{2d}$. Across $d$ coordinate, this number is bounded by $O\left(\frac{d^2\left(\log\log\frac{M}{m} + \log\frac{d}{\beta}\right)\log\frac{M}{m}}{\varepsilon}\right)$ with probability $\geq 1 - \frac{\beta}{2}$. $\square$

**Privacy guarantee.**

**Lemma 24.** *Algorithm 4 is $(\varepsilon, \delta)$-DP.*

*Proof.* The pre-processing step is $\frac{\varepsilon}{2}$-DP by Lemma 23. In each iteration, we use the Exponential mechanism with privacy parameter $\varepsilon'$ to find $\Delta_t$. By strong composition over $T$ iterations, Algorithm 4 is $\left(\varepsilon'\sqrt{2T\log\frac{1}{\delta}} + \frac{T\varepsilon'^2}{2}, \delta\right)$-DP. For $\varepsilon' = \frac{\varepsilon}{4\sqrt{T\log\frac{1}{\delta}}}$ and constant $\varepsilon$, this implies Algorithm 4 is $(\varepsilon, \delta)$-DP. $\square$

**Utility Analysis.**

**Lemma 25.** *With probability at least $1 - \beta$, for all $t \in [T]$:*

$$\frac{\left\langle \nabla smax^U\left(Ax_{t-1}\right), P\Delta_t\right\rangle}{\left\langle \nabla smin^U\left(Ax_{t-1}\right), C\Delta_t\right\rangle} \leq 1 + \frac{\alpha}{5}.$$

*Claim* 26. There exists $y \geq 0$ such that $\sum_i M_i x_i \leq d$, $Py \leq 1 + \frac{\alpha}{20}$ and $Cy \geq \frac{1}{1 + \alpha/20}$.

*Proof.* Let $x^*$ be such that $P^{\text{init}} x^* \leq 1$ and $C^{\text{init}} x \geq 1$.

Let $y$ be such that $y_j = \frac{1}{1 + \alpha/20}\left(x_j^* + \frac{\alpha}{20 d M_j}\right)$. We have $\sum_j M_j y_j \leq \frac{1}{1 + \alpha/20}\left(d + \frac{\alpha}{20}\right) \leq d$.

Consider packing constraint $i$.

$$P_i y \leq \frac{1}{1 + \alpha/20}\left(1 + \frac{\alpha}{20}\right) = 1.$$

Consider covering constraint $i$.

If there is some $j$ such that $C_{ij}^{\text{init}} \geq \frac{40 d M_j}{\alpha}$, we have $C_i y \geq \frac{1}{1 + \alpha/20}\frac{40 d M_j}{\alpha} \cdot \frac{\alpha}{20 d M_j} \geq 1$. Otherwise we have $C_i \geq C_i^{\text{init}}$. Hence

$$C_i y = C_i^{\text{init}} y \geq C_i^{\text{init}}\frac{x^*}{1 + \alpha/20} \geq \frac{1}{1 + \alpha/20}.$$

$\square$

*Proof.* We calculate the sensitivity of the score function $Q(j) = -\frac{\left\langle \nabla smax^U\left(Px\right), P1_j\right\rangle}{\left\langle \nabla smin^U\left(Cx\right), C1_j\right\rangle}$. We consider two cases: when the neighboring inputs differ by a covering constraint and when the differ by a packing constraint.

Case 1: Suppose that neighboring inputs are given by $(P, C)$ and $(P, C')$. Note that after pre-processing, we have $C_{\min}$ and $C'_{\min}$ are bounded from below by $\frac{\alpha M_j}{d}$. Thus we have

$$\left| \frac{\left\langle \nabla \mathrm{smax}^U \left( Px \right), P1_j \right\rangle}{\left\langle \nabla \mathrm{smin}^U \left( Cx \right), C1_j \right\rangle} - \frac{\left\langle \nabla \mathrm{smax}^U \left( Px \right), P1_j \right\rangle}{\left\langle \nabla \mathrm{smin}^U \left( C'x \right), C'1_j \right\rangle} \right|$$

$$= \left| \left\langle \nabla \mathrm{smax}^U \left( Px \right), P1_j \right\rangle \right| \frac{\left| \left\langle \nabla \mathrm{smin}^U \left( Cx \right), C1_j \right\rangle - \left\langle \nabla \mathrm{smin}^U \left( C'x \right), C'1_j \right\rangle \right|}{\left\langle \nabla \mathrm{smin}^U \left( Cx \right), C1_j \right\rangle \left\langle \nabla \mathrm{smin}^U \left( C'x \right), C'1_j \right\rangle}$$

$$\leq \frac{M_j}{s} \frac{3 \frac{40 d M_j}{\alpha}}{\left( \frac{\alpha M_j}{d} \right)^2} = \frac{120 d^2}{s \alpha^3}.$$

Case 2: Suppose that neighboring inputs are given by $(P, C)$ and $(P', C)$. We also have

$$\left| \frac{\left\langle \nabla \mathrm{smax}^U \left( Px \right), P1_j \right\rangle}{\left\langle \nabla \mathrm{smin}^U \left( Cx \right), C1_j \right\rangle} - \frac{\left\langle \nabla \mathrm{smax}^U \left( P'x \right), P'1_j \right\rangle}{\left\langle \nabla \mathrm{smin}^U \left( Cx \right), C1_j \right\rangle} \right|$$

$$\leq \frac{3 M_j}{s \left( \frac{\alpha M_j}{d} \right)} \leq \frac{3d}{s\alpha} \leq \frac{120 d^2}{s \alpha^3}.$$

There must exist a coordinate $j$ such that

$$\frac{\left\langle \nabla \mathrm{smax}^U \left( Px_{t-1} \right), P1_j \right\rangle}{\left\langle \nabla \mathrm{smin}^U \left( Cx_{t-1} \right), C1_j \right\rangle} \leq \left( 1 + \frac{\alpha}{20} \right)^2.$$

Therefore with probability at least $1 - \frac{\beta}{T}$,

$$\frac{\left\langle \nabla \mathrm{smax}^U \left( Px_{t-1} \right), P1_j \right\rangle}{\left\langle \nabla \mathrm{smin}^U \left( Cx_{t-1} \right), C1_j \right\rangle} \leq \left( 1 + \frac{\alpha}{20} \right)^2 + \frac{120 d^2 \left( \log d + \log \frac{T}{\beta} \right)}{s \alpha^3 \varepsilon'} \leq 1 + \frac{\alpha}{5}.$$

$\square$

**Lemma 27.** *With probability $1 - \beta$, the output $\overline{x}$ of Algorithm 3 satisfies $P_i \overline{x} \leq 1 + \alpha$ and $C_i \overline{x} \geq 1 - \alpha$ except for at most $r$ constraints where*

$$r = O \left( \frac{d^3 \left( \log d + \log \frac{n}{\alpha \beta} \right) \sqrt{\log \frac{1}{\delta} \log n}}{\alpha^6 \varepsilon} + \frac{d^2 \left( \log \log \frac{M}{m} + \log \frac{d}{\beta} \right) \log \frac{M}{m}}{\varepsilon} \right).$$

*Proof.* The proof is similar to the that of Lemma 21. The number of violated constraints is

$$s = O \left( \frac{d^2 \left( \log d + \log \frac{T}{\beta} \right)}{\alpha^4 \varepsilon'} \right) = O \left( \frac{d^2 \left( \log d + \log \frac{T}{\beta} \right) \sqrt{T \log \frac{1}{\delta}}}{\alpha^4 \varepsilon} \right)$$

$$= O \left( \frac{d^3 \left( \log d + \log \frac{n}{\alpha \beta} \right) \sqrt{\log \frac{1}{\delta} \log n}}{\alpha^6 \varepsilon} \right).$$

Taken into account the number of filtered constraints, we obtain the bound in the lemma. $\square$

