# OpenReview forum: "Solving Positive Linear Programs with Differential Privacy"
_ICML.cc/2026/Conference — ICML 2026 regular_

### Official Review · Reviewer_nEP1 · 2026-03-09

**Soundness:** 4
**Presentation:** 4
**Significance:** 3
**Originality:** 3
**Overall Recommendation:** 5
**Confidence:** 4

**Summary:**

This paper studies solving positive LPs (LPs with no negative coefficients or variables, i.e., packing LPs, covering LPs, and mixed packing-covering LPs) with $(\epsilon, \delta)$-differential privacy.  The particular notion of DP used is that two positive LPs are neighbors if they differ in exactly one constraint (but this constraint can differ essentially arbitrarily).  Due to this definition of LP, by necessity some constraints will need to be violated by any DP algorithm, so the goal is to get a good approximation of the optimal solution while minimizing the number of violated constraints.  Previous work studies either special cases (e.g., fractional set cover) or solving general LPs.  By focusing particularly on positive LPs, the authors are able to take the same basic idea used in the first paper on DP LP solving (Hsu et al), in particular, dense multiplicative weights, but show how to leverage the special structure of positive LPs to get notably improved bounds.  All of their bounds come in two flavors: data-dependent bounds, which depend on things like the maximum coefficient or the optimal value itself, and data-independent bounds, which depend only on the dimension.

**Compliance With Llm Reviewing Policy:**

Affirmed.

**Final Justification:**

The rebuttal addressed the main weakness I pointed out in my review, and so essentially reinforced my prior assessment that this is a very strong paper that I think should be accepted.

**Key Questions For Authors:**

- What would be a setting where this problem is well-motivated but the bounds you get are actually reasonable?

**Limitations:**

yes

**Strengths And Weaknesses:**

### Strengths:
- The bounds seem notably stronger than related work, although as they point out, this is because a) they are only handling positive LPs, and b) they are in the "low-precision" setting where they're allowed to depend linearly on the approximation rather than logarithmically.  But the results still seem like a notable improvement to me.
- While in some sense their techniques are similar to Hsu et al., they have a very different approach and analysis in order to get improved bounds for positive LPs.  I think the technical difficulty is well above the bar.
- I completely agree with the authors that positive LPs are an extremely interesting class of LPs for which it makes sense to design specialized algorithms.
- The presentation is quite good -- I found this a reasonably easy to read paper.

### Weaknesses:
- The only real weakness in my mind is that I'm not sure how well-motivated this whole line of work is.  In particular, while the bounds here are notably stronger than previous work, they still require us to violate a huge number of constraints (something like $n^{3/2}$ where $n$ is the number of variables in the LP).  That seems like a lot to me.  If you look at the examples they use on page 1 to motivate positive LPs, it seems to me like in all of them, violating $n^{3/2}$ constraints arbitrarily would actually be completely useless.  It would have been nice if the authors presented even a single example of a natural setting where one might want to solve a positive LP privately and yet still be OK with the constraint violations that they get.  I can't think of anything, so I somewhat suspect that no one would ever be willing to use any of these techniques since requiring DP in this setting makes the algorithms useless.  But to be fair, there is significant history of this problem and it is inherently interesting mathematically, so I still think that this paper should be accepted.

---

> ### Author Rebuttal · Authors · 2026-03-30
>
> We thank the reviewer for the valuable feedback. We answer the reviewer's concern and questions below.
>
> In our paper, beyond our new input-independent results, we also provide improved instance-specific bounds compared to prior work. For example, for covering LPs, the bound is $\tilde{O}\left(\frac{\left(A_{\max}OPT\right)^{1.5}}{\alpha^{2}\epsilon}\right)$. Depending on problems, this bound can be smaller than the input-independent bound $\tilde{O}\left(\frac{d^{1.5}}{\alpha^{3.5}\epsilon}\right)$.
>
> The bound $\tilde{O}(d^{1.5})$ is reasonable is when $d\ll n$ i.e. there are much fewer variables than the number of users. An example is setting the prices (the variables) of some common goods among a large user base to maximize revenue while making sure that most users can meet their needs. Here, the number of users $n$ is vast, while the number of variables d remains small. This is a natural and well-motivated setting where a $d$-dependent bound is reasonable. Another example is the combinatorial public project problem where the goal is to select $k$ public projects from a set of $d$ candidates, so as to maximize the total utility of $n$ users where $n\gg d$. Each project can benefit different users differently and benefit a lot of users e.g. funding a health clinic can serve a large number of users living near the clinic but not users far away. This is a discrete problem but an LP relaxation can still yield valuable information such as by rounding.

---

> > ### Author Rebuttal · Reviewer_nEP1 · 2026-04-01
> >
> > These are good examples.  I will keep my score at 5.

---

### Official Review · Reviewer_EWXG · 2026-03-11

**Soundness:** 4
**Presentation:** 3
**Significance:** 3
**Originality:** 3
**Overall Recommendation:** 4
**Confidence:** 4

**Summary:**

The authors introduce differentially private approximation algorithms for solving linear programs with non-negative coefficients and variables under the high-sensitivity constraint private setting (where neighboring LP's may differ by a single arbitrary constraint).

They provide algorithms that approximately satisfy all but a bounded number of constraints. The number of violated constraints is bounded by either the problem dimension or entries of the LP, which improve upon current results in the number of constraints violated.

**Compliance With Llm Reviewing Policy:**

Affirmed.

**Final Justification:**

The rebuttal clarified my main point. Overall, I continue to view the summary contributions as above-bar for ICML but not "slam dunk": hence my continued score of "Weak Accept".

**Key Questions For Authors:**

In Lemma 17, since Algorithm 2 describes a covering LP, the output should satisfy $A_i \bar{x}\geq 1-\alpha$.

**Limitations:**

Yes

**Strengths And Weaknesses:**

### Soundness
The paper supports its theoretical claims with sound proofs. The authors assume that the non-private LP optimum is known, citing previous work which also uses this assumption. In addition, the authors also assume that entries of the LP come from known bounded ranges. This may be justified for some applications of LP such as set cover (where the ranges are fixed for all instances of the problem), additional discussion may be warranted for other problems where these assumptions may incur privacy loss.

### Presentation
The authors include descriptions of their algorithms and explanations on their design choices to aid the reader's understanding. The proof of main theorems are partitioned into well-chosen lemmas. Overall relatively well-presented.

### Significance
This paper addresses an important topic on privacy and optimization. Since LP is a foundational tool in machine learning, data-driven decision systems, and many other algorithms, obtaining private but non-trivial solutions is highly relevant. Even though applicable to only positive LP's, the results may pave the way for future research on private optimization, and may find uses in sensitive real world domains.

### Originality
This work focuses on the high-sensitivity constraint-private regime introduced by Hsu et al. (https://doi.org/10.48550/arXiv.1402.3631). The contributions are distinct from the related works in that the authors provided asymptotically better bounds. Although another work (https://doi.org/10.48550/arXiv.2501.19315) has also explored differentially private LP without violating constraints under a slightly different but comparable privacy model (privacy on the underlying dataset).

[^1]: [https://doi.org/10.48550/arXiv.1402.3631](https://doi.org/10.48550/arXiv.1402.3631)
[^2]: [https://doi.org/10.48550/arXiv.2501.19315](https://doi.org/10.48550/arXiv.2501.19315)

---

> ### Author Rebuttal · Authors · 2026-03-30
>
> We thank the reviewer for the valuable feedback. Indeed there was a typo in Lemma 17, which we will correct. We are happy to answer any follow up questions.

---

> > ### Author Rebuttal · Reviewer_EWXG · 2026-04-02
> >
> > The single concern is resolved.

---

### Official Review · Reviewer_b4Lr · 2026-03-12

**Soundness:** 3
**Presentation:** 2
**Significance:** 3
**Originality:** 3
**Overall Recommendation:** 4
**Confidence:** 4

**Summary:**

// problems
This paper studies solving positive (packing, covering, and mixed packing-covering) linear programs with DP.
Specifically, for packing LPs (covering and mixed LPs are similar), an input matrix $A \ge 0_{n \times d}$ defines the following LP: $\max 1^\top x$ s.t. $x \ge 0$ and $Ax \le 1$,
$A$ and $A'$ are neighboring if they differ by a single row.

//results:
Given a packing LP instance, the $(\epsilon, \delta)$-DP algorithm given by this paper outputs a solution $x$ satisfies $Ax \le 1 + \alpha$ except for at most $\tilde{O}(\frac{(A_\max OPT)^{1.5}}{\alpha^2 \epsilon})$ or $\tilde{O}(\frac{d^{1.5}}{\alpha^{3.5}\epsilon})$ constraints.
They also give similar results for covering LPs and mixed LPs.

//technique:
Their algorithmic technique is built on dense MWU for positive LPs. (The general framework is proposed by Hsu et al. (2014))

**Compliance With Llm Reviewing Policy:**

Affirmed.

**Final Justification:**

The rebuttal partially addressed my main concerns. The problem and results in this paper look interesting, but I would suggest improving the presentation. Overall, I think this paper is on the borderline of acceptance.

**Key Questions For Authors:**

- On page 2, in the paragraph of "Our technique", you wrote "However, this blackbox approach fails to leverage the positivity of the constraints. Instead, we develop our new toolset from first principles, which allow us to exploit the structure of the problem." Could you explain in more detail how previous blackbox approach fails to leverage the constraint positivity, how your algorithm is able to take advantage, and what improvements you get as a result?
- I'm a bit confused by the neighboring relation in DP. Is your algorithm private for any A, A' that differ by at most row? Or only for positive A, A' that differ by at most row?
- It seems the results of Kaplan et al. (2024) and Ene et al. (2025) show that one can exactly (instead of approximately) satisfy all except some number of constraints. In this paper, your algorithms only approximately satisfy all but some number of constraints. Can you say more about the difference? (Of course, one way to exactly satisfy those constraints is to simply scale x, but I'm not sure if this is the right way.)

I would be happy to increase my score if my questions are addressed properly

**Limitations:**

yes

**Strengths And Weaknesses:**

// strengths
- Solving LPs with DP is an important problem in the DP literature, and positive LPs are an important subclass of LPs. So it makes great sense to investigate if improvements could be made, compared with general LPs, when tailored to this subclass.
- The use of truncated softmax and softmin seems new, compared with prior works.

// weakness
- It looks to me some assumptions in the paper, e.g. OPT is known and $b = 1, c = 1$, are made to better communicate the idea or to simplify the notation. But I think there should also be some general statements, at least more discussions, without making these assumptions so that it would be easier for the readers to see those explicit dependencies.

---

> ### Author Rebuttal · Authors · 2026-03-30
>
> We thank the reviewer for the valuable feedback. We address the reviewer's questions below.
>
> Q1. How our approach leverages constraint positivity.
>
> In short, our algorithm uses constraint positivity to make larger steps when updating the solution. For the analysis, we need to design a new potential function to exploit the constraint positivity, which is unclear how to do with the previous approach for general LPs. More specifically, we look at the $\mathrm{smax}^{U}$ function which we use to track the quality of the solution we build iteratively. Without the constraint positivity, when we update $x\to x+\Delta$ , we track the change in the potential function of the constraints $$\mathrm{smax}^{U}(A(x+\Delta))-\mathrm{smax}^{U}(Ax)$$
>
> We can bound $$\mathrm{smax}^{U}(A(x+\Delta))-\mathrm{smax}^{U}(Ax)	\lesssim\langle\nabla\mathrm{smax}^{U}(Ax),A\Delta\rangle+||A\Delta||\_{\infty}^{2}$$
>
> The $||A\Delta||\_{\infty}^{2}$ term results in the convergence rate $\frac{\rho^{2}\log n}{\alpha^{2}}$ of Dense MWU for general LP, where $\rho$  is the width of the problem, being an upper bound for $||A\Delta||\_{\infty}$. On the other hand, if we have the constraint positivity, we can bound $$\mathrm{smax}^{U}(A(x+\Delta))-\mathrm{smax}^{U}(Ax)	\lesssim(1+||A\Delta||\_{\infty})\langle\nabla\mathrm{smax}^{U}(Ax),A\Delta\rangle$$
>  and improve the convergence rate to $\frac{\rho\log n}{\alpha^{2}}$.
>
> The improvement of the runtime of the non-private algorithm by a factor $\rho$  leads to the improvement in the number of violated constraints in our private algorithm. Intuitively every iteration requires some privacy loss so having fewer iterations allows us to utilize the privacy budget more efficiently.
>
> 2. Definition of DP
>
> Here neighboring inputs are defined as positive matrices $A$, $A'$ that differ by at most one row.
>
> 3. Comparison with Kaplan et al. (2024) and Ene et al. (2025)
>
> For pure LPs (packing and covering), we can obtain a solution that satisfies most constraints exactly by appropriately scaling the approximate solution as the reviewer suggested. This rescaling preserves the $(1\pm O(\alpha))$-approximation of the OPT value. We also refer the reviewer to our [response to Reviewer w8mf](https://openreview.net/forum?id=zlSioMUQ2Y&noteId=q4SraJR3VK) for a more detailed comparison between our work and Kaplan et al. (2024) and Ene et al. (2025).

---

> > ### Author Rebuttal · Reviewer_b4Lr · 2026-04-01
> >
> > Thanks for the response. I would still like to see some explicit dependence of the final utility bound on OPT, b and c. I saw the authors pointed out the dependence on OPT, but it would be good to see a full-fledged statement.

---

> > > ### Author Response · Authors · 2026-04-06
> > >
> > > Thank you for the response.
> > >
> > > We wrote in the paper that we assume $b=1$ and $c=1$ w.l.o.g and further explanation is as follows. Since the objective function $c^{\top}x$ is public, we can change the variable $x'\_{i}=x\_{i}c\_{i}$ and use the objective function $\sum\_{i}x'\_{i}$ instead. All users can do the substitution by themselves for their own constraint. Each user $j$ can also normalize their own constraint so that the RHS $b_{j}=1$. For example, $2x_{1}+4x_{2}\le2$ can be rewritten equivalently as $x_{1}+2x_{2}\le1$.
> > >
> > > Our argument for the packing case (similarly for the other cases) can be made explicitly as follows. We assume that the range of $\left\Vert A_{:,i}\right\Vert _{\infty}$ (the maximum coefficient of each variable) is $[L,U]$. We can derive that OPT lies in the range $[\frac{1}{dU},\frac{1}{L}]$. By doing a binary search for OPT over this range and using the exponential mechanism to find the best guess, we obtain that the bound is $\tilde{O}\left(\frac{d^{3/2}}{\alpha^{3.5}\epsilon}+\frac{\log\log\frac{dU}{L}}{\epsilon}\right)$.

---

### Official Review · Reviewer_w8mf · 2026-03-14

**Soundness:** 4
**Presentation:** 4
**Significance:** 3
**Originality:** 4
**Overall Recommendation:** 4
**Confidence:** 3

**Summary:**

The paper studies solving linear programs while keeping the parameters differentially private. They follow the setup of Hsu-Roth-Roughgarden-Ullman (ICALP 2014), and proposes a method with new data-independent bounds that improves the prior theoretical guarantees.

**Compliance With Llm Reviewing Policy:**

Affirmed.

**Ethical Review Concerns:**

None.

**Final Justification:**

My questions are addressed. I keep my positive score of 4.

**Key Questions For Authors:**

1. If the methods of Kaplan et al. (2024) and Ene et al. (2025) are applied specifically to packing, covering, and mixed packing-covering LPs, is it possible to improve their original bounds for general LPs?
2. Do Kaplan et al. (2024) and Ene et al. (2025) make the assumption that OPT is known?

**Limitations:**

Yes.

**Strengths And Weaknesses:**

**Strengths**
- The paper investigates an important and challenging problem. Linear programming encompasses a wide range of practical problems and maintaining differential privacy while solving the LP is fundamentally challenging.
- The paper proposes a novel method that employs the Dense Multiplicative Weights Update algorithm and updates the constraint weights by more carefully picked values.
- The method is able to improve the previous methods for three specific types of LPs: packing, covering, and mixed packing-covering LPs, where parameters b and c are constantly 1.

**Weaknesses**
- The improvements compared to prior work seem to have tradeoffs.
    - Kaplan et al. (2024) and Ene et al. (2025) seem to focus on general LPs, while this work considers three specific types of LPs.
    - The proposed method seems to assume that the optimal objective value of the LP is known. I did not find discussion regarding whether Kaplan et al. (2024) and Ene et al. (2025) made the same assumption. This seems to be a quite strong assumption.

---

> ### Author Rebuttal · Authors · 2026-03-30
>
> We thank the reviewer for the valuable feedback. We answer the reviewer's questions below.
>
> Q1. If the goal is to find a $(1\pm\alpha)$-approximate solution, Kaplan et al. (2024) and Ene et al. (2025) focus on the "high-precision" regime seeking dependencies of poly$\log\frac{1}{\alpha}$; whereas we operate in the "low-precision" regime, allowing for poly$\frac{1}{\alpha}$ dependence. When privacy is not imposed, algorithms for the low precision regime have better runtime. When privacy is imposed, a key contribution of our work is the improved dependence on the problem dimension in the number of violated constraints. While Ene et al. (2025)'s algorithm violated $O(d^{4})$ constraints, our algorithms reduce this to $d^{3/2}$.
>
> The difference in the approximation regimes requires different algorithmic techniques. In the low-precision regime, we employ a custom version of Dense MWU, which allows us to obtain improved bounds for positive LPs. The algorithms in high-precision regime instead are based on variants of perceptron, which is applicable to general LPs, but to our knowledge, no improvement for positive LPs is known.
>
> Q2. The assumption that OPT is known (also seen in Hsu et al., 2014) was adopted primarily to simplify the presentation and is not a fundamental limitation of our framework. To lift this assumption, one can employ a standard binary search over the range of OPT, and use the exponential mechanism to identify the optimal guess. This extension would only introduce a marginal overhead of a $\log$(Range of OPT) factor to the final utility bound. We will clarify this in the final version of the manuscript.
>
> In Kaplan et al. (2024) and Ene et al. (2025), the primary goal is to find a feasible solution to the system $Ax\le b, x\ge0$, without an objective. The standard reduction from an optimization problem to a feasibility problem inherently requires a binary search over the objective value. Consequently, our approach does not rely on stronger assumptions than those employed in prior work.

---

> > ### Author Rebuttal · Reviewer_w8mf · 2026-04-02
> >
> > Thank you for the explanations.

---

### Decision · Program_Chairs · 2026-04-30

**Decision:**

Accept (regular)

**Comment:**

Reviewers agreed that the paper considers a reasonable problem and clearly presents a meaningful technical contribution to solve it. The authors should make sure to incorporate the discussion feedback, particularly with respect to the problem regimes where the paper's contribution is most useful.